# DYNAMIC DEMONSTRATIONS CONTROLLER FOR IN-CONTEXT LEARNING

## ABSTRACT

In-Context Learning (ICL) is a new paradigm for natural language processing (NLP), where a large language model (LLM) observes a small number of demonstrations and a test instance as its input, and directly makes predictions without updating model parameters. Previous studies have revealed that ICL is sensitive to the selection and the ordering of demonstrations. However, there are few studies regarding the impact of the demonstration number on the ICL performance within a limited input length of LLM, because it is commonly believed that the number of demonstrations is positively correlated with model performance. In this paper, we found this conclusion does not always hold true. Through pilot experiments, we discover that increasing the number of demonstrations does not necessarily lead to improved performance. Building upon this insight, we propose a ***Dynamic Demonstrations Controller*** (***$D^2$Controller***), which can improve the ICL performance by adjusting the number of demonstrations dynamically. The experimental results show that $D^2$Controller yields a 5.4% relative improvement on eight different sizes of LLMs across ten datasets. Moreover, we also extend our method to previous ICL models and achieve competitive results.

## 1 INTRODUCTION

In-Context Learning (ICL) is a new paradigm for performing various NLP tasks using large language models (LLMs) (Brown et al., 2020). In ICL, by conditioning on a small number of *demonstrations*, LLMs can generate predictions for a given test input without updating model parameters. Restricted by the maximum input length of LLMs, it is common to sample a small set of examples from the training dataset randomly to formulate demonstrations. Figure 1 shows an example of sentiment analysis using ICL.

To improve the performance of ICL, existing work primarily focuses on designing *Demonstration Selection* methods (Liu et al., 2022a; Rubin et al., 2022; Zhang et al., 2022b; Kim et al., 2022; Gonen et al., 2022; Sorensen et al., 2022; Wang et al., 2023; Li et al., 2023; Li & Qiu, 2023) or finding an appropriate *Demonstration Ordering* (Lu et al., 2022; Wu et al., 2022), since a lot of studies have revealed that ICL is sensitive to the selection as well as the ordering of demonstrations (Liu et al., 2022a; Rubin et al., 2022; Zhang et al., 2022b; Lu et al., 2022; Wu et al., 2022; Li et al., 2023; Li & Qiu, 2023; Dong et al., 2022).

However, to the best of our knowledge, there are few studies available regarding the impact of the *Demonstration Number* on the ICL performance. This scarcity may be attributed to the prevailing belief that the relation between the number of demonstrations and model performance follows a power law – as the number of demonstrations increases, model performance continues to improve (Xie et al., 2022; Xu et al., 2023). Nevertheless, through pilot experiments, we find this conclusion does not always hold true. Specifically, within the constraints of input length in LLMs, we systematically evaluate model performance across a spectrum ranging from the minimum to the maximum number of demonstrations. This comprehensive assessment involves five different datasets and encompasses five sizes of LLMs (Brown et al., 2020; Zhang et al., 2022a; Dey et al., 2023). Our findings reveal that:

- As more demonstrations are incorporated into the model input, the changes of the performance across different datasets on the same model tend to be inconsistent, with some

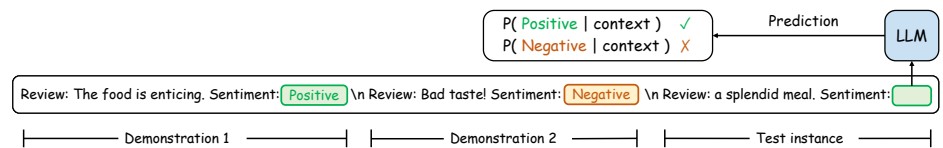

Figure 1: An example of In-Context Learning. ICL takes a small number of demonstrations and a test instance as input, with large language model responsible for making predictions.

datasets showing improvements while others experiencing declines. Similarly, the performance of different models on the same dataset also rises or falls. This suggests that increasing the number of demonstrations does not necessarily improve performance.

- During the transition from minimum to maximum number of demonstrations, the number of demonstrations needed for the same model to attain the optimal performance varies across different datasets. Likewise, different models exhibit variations in the number of demonstrations required to reach the optimal performance on the same dataset. This suggests that the optimal number of demonstrations may differ depending on the specific dataset and model combination.

Based on the above observation, we can infer that it is necessary to dynamically select an appropriate demonstration number for different datasets and models. Doing so not only boosts ICL performance but also can help in saving time and space during the inference of LLMs. To achieve this goal, we propose a ***Dynamic Demonstrations Controller*** (***D²Controller***), the core idea of which involves comparing the prediction accuracy of different demonstration numbers on a small set of specially selected evaluation examples. The key challenge of this idea is determining which evaluation examples should be chosen to provide a correct assessment for different demonstration numbers. To tackle this challenge, we design a metric named ***Intra-Inter-Class Score*** (***IICScore***) to guide the D²Controller to select suitable evaluation examples from the training dataset. Finally, we apply D²Controller to eight different sizes of LLMs and achieve a 5.4% relative improvement over ten datasets. Besides, we also extend our method to previous ICL models and achieve competitive results.

Our contributions are summarized as follows: (1) We comprehensively analyze the effects of the number of demonstrations on ICL performance under a limited input length of LLM, and find that the number of demonstrations may not necessarily be positively correlated with model performance; (2) We propose a method named D²Controller, which not only boosts ICL performance but also saves time and space during inference of the LLMs; (3) We apply our method to eight different sizes of LLMs and realize an average of 5.4% relative improvement across ten datasets. Moreover, we also extend our method to previous ICL models and yield competitive results.

## 2 BACKGROUND

In this section, we review the definition of In-Context Learning and the $k$-shot setting.

**Notation**   We use $\boldsymbol{\theta}$ to denote an LLM. The training dataset is denoted as $\mathcal{D}$. A training example $(x_i, y_i)$ consists of a sentence $x_i$ and a label $y_i$. The sentence of a training example is also referred to as an *instance*. We use $\mathcal{I}_\mathcal{D} = \{x_i\}_{i=1}^{|\mathcal{D}|}$ to represent all instances of training examples in $\mathcal{D}$. The label space is denoted as $\mathcal{Y}$. In this paper, we focus on ICL for text classification tasks. Each training example belongs to a certain class. The set of classes is represented as $\mathcal{C}$ and a class $c \in \mathcal{C}$ has a one-to-one correspondence with a label $y^c \in \mathcal{Y}$, *i.e.*, $|\mathcal{Y}| = |\mathcal{C}|$. For example, the label "not entailment" corresponds to the class in which premise sentences do not entail hypothesis sentences.

### 2.1 IN-CONTEXT LEARNING

Given an LLM $\boldsymbol{\theta}$, a group of $n$ in-context examples $\{x_i, y_i\}_{i=1}^n$ sampled from training dataset $\mathcal{D}$ (In general, $n \ll |\mathcal{D}|$), and a test instance $x_\text{test}$, ICL first formulates in-context examples in the format of the input-label pairs which are named the *demonstrations* (See Appendix A for details)

via templates, and then concatenates them together along with a test input to construct a prompt $P$:

$$P = \Omega(x_1, y_1) \oplus \Omega(x_2, y_2) \oplus \cdots \oplus \Omega(x_n, y_n) \oplus \Omega(x_{\text{test}}, *), \tag{1}$$

where $\Omega(\cdot, \cdot)$ denotes template-based transformation and $\oplus$ means concatenation operation. Notice that there is a verbalization process $\pi(\cdot)$ inside $\Omega(\cdot, \cdot)$, which maps the label $y_i$ to a token $v_i$ in the LLM vocabulary. The $y_i$ and $v_i$ can be different. For example, the label "not entailment" can be mapped to the token "false". We denote the mapping token space as $\mathcal{V}$ and we have $|\mathcal{Y}| = |\mathcal{V}|$ (See Appendix A for details). Finally, The prompt $P$ is fed into the LLM $\boldsymbol{\theta}$ to predict the label of the test instance $x_{\text{test}}$:

$$\hat{y}_{\text{test}} = \pi^{-1}(\arg\max_{v \in \mathcal{V}} \quad \boldsymbol{p}(v|P, \boldsymbol{\theta})), \tag{2}$$

where $\pi(\cdot)^{-1}$ denotes the inverse mapping from the token $v_i$ to the label $y_i$.

## 2.2 $k$-SHOT SETTING

For text classification tasks, each prompt $P$ is formulated in the class balance way, *i.e.*, the demonstrations of each class are contained in a prompt $P$ and the numbers of them are the same[1]. Among them, the number of demonstrations of each class is also called the *shot number*, denoted as $k$. Based on this, the $k$-shot setting means a prompt $P$ contains $k$ demonstrations for each class. In other words, the total demonstration number $n$ of each prompt $P$ is equal to $k|\mathcal{C}|$. In this paper, we vary the number of demonstrations $n$ by changing the $k$-shot setting.

Due to the input length limitation of LLMs, there exists a maximum $k$, denoted as $k_{\max}$, for every dataset. All feasible choices of $k$ for a dataset form a set $\mathcal{K} = \{1, 2, \cdots, k_{\max}\}$ (Appendix B provides the calculation method for $k_{\max}$ and the value of $k_{\max}$ for each dataset).

## 3 PILOT EXPERIMENTS

In this section, we conduct pilot studies to answer the following research question: *Does model performance consistently improve when more demonstrations are added to prompts?*

**Experimental Setup**   We conduct pilot experiments across five text classification datasets on five different sizes of LLMs, including two Cerebras-GPT models (Dey et al., 2023) (with 2.7B and 6.7B parameters), two OPT models (Zhang et al., 2022a) (with 13B and 30B parameters) and a GPT-3 model (Brown et al., 2020) (with 175B parameters). We adopt *Accuracy* as the evaluation metric for model performance (Lu et al., 2022; Zhang et al., 2022b). Following (Lu et al., 2022; Xu et al., 2023), we randomly sample 256 examples from the validation set for each dataset to evaluate the accuracy and report the average performance and standard deviation based on 5 different seeds.

For each dataset, we iteratively test the model performance from 1-shot setting to $k_{\max}$-shot setting on five sizes of LLMs. Figure 2 and Figure 3 show the performance curves of five datasets on Cerebras-GPT 6.7B model and GPT-3 175B model, respectively. Figure 4 shows performance curves of the SST5 dataset on five different sizes of LLMs. More results are provided in Appendix C and Appendix F. Based on these results, we find that:

**Increasing the number of demonstrations does not necessarily improve the model performance.**   In Figure 2, we can see that when more demonstrations are added to prompts, *i.e.*, the shot number is increased, the model performance goes up or down on five different datasets. From a local point of view, when changing from an 8-shot setting to a 16-shot setting on the MPQA dataset, the model performance increases from 71.5 to 83.1, while the accuracy drops to 79.8 with a 32-shot setting. Likewise, on the CB dataset, the accuracy declines when shifting from a 2-shot setting to a 4-shot setting. Furthermore, when providing more demonstrations on the SST-5 dataset, the model's performance consistently decreases. From the perspective of a general trend, the accuracy improves on the MPQA dataset while declines on the CB and SST-5 datasets. Similar observations can be found in the results of the GPT-3 175B model, shown in Figure 3. Besides, the performance of different models on the same dataset also rises or falls. As shown in Figure 4, when changing from a

---

[1]For example, in a 2-class sentiment analysis task, a prompt $P$ contains demonstrations from both the positive sentiment class and the negative sentiment class.

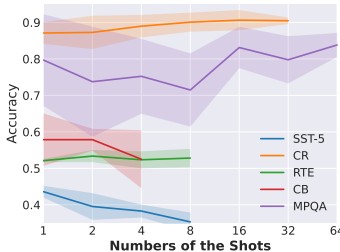 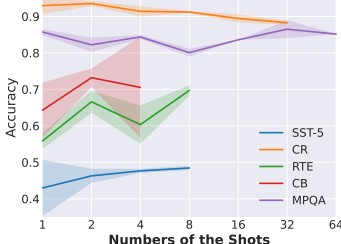 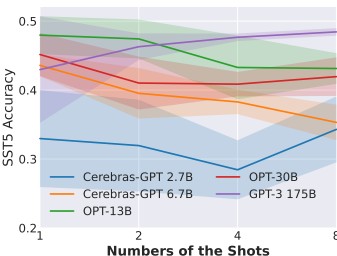

Figure 2: Effect of the demonstrations number on Cerebras-GPT-6.7B across five datasets.

Figure 3: Effect of the number of demonstrations on GPT-3 175B across five datasets.

Figure 4: The accuracy of five different sizes of LLMs on the SST5 dataset.

1-shot setting to a 8-shot setting, the accuracy of the SST5 dataset on the OPT-13B model continues to decrease, while that on the GPT-3-175B model keeps rising. These observations indicate that the inclusion of more demonstrations does not guarantee improved performance.

**The optimal $k$-shot setting differs depending on specific datasets and models.** Here we define the $k$-shot setting under which a dataset acquires the highest accuracy as the optimal $k$-shot setting. From Figure 3, we can tell that the optimal $k$-shot setting for each dataset is different: 2-shot setting for CR and CB datasets, 8-shot setting for RTE and SST5 dataset and 32-shot setting for MPAQ dataset. Jointly observing Figure 2 and Figure 3, we find that the optimal $k$-shot settings for the same dataset on different models can be different. The curves in Figure 4 further support this finding.

From the above analysis, we can infer that to achieve better performance in ICL, it is not appropriate to simply use the $k_{\max}$-shot setting for each dataset or the same $k$-shot setting for all datasets. The latter is a strategy widely adopted in previous work (Lu et al., 2022; Xu et al., 2023). Instead, we should dynamically decide $k$-shot settings for ICL depending on specific datasets and models.

## 4 METHODOLOGY

Based on the observations of the pilot study, we propose a **Dynamic Demonstrations Controller** (**$D^2$Controller**), which dynamically finds a suitable $k$ from the feasible shot numbers set $\mathcal{K}$ for each dataset. An intuitive way to decide an appropriate $k$ for a specific dataset is to compare the average prediction accuracy of different $k$-shot settings on a set of evaluation examples and make a choice. The key challenge of such idea lies in that on which evaluation examples we can obtain the proper evaluation for each $k$-shot setting.

To tackle the above challenge, we propose a metric named **Intra-Inter-Class Score** (**IICScore**) to guide us to choose the representative evaluation examples for each group of in-context examples from the training dataset. The whole process to evaluate each $k$-shot setting is divided into three steps: (1) In-context examples sampling. (2) IICScore-guided evaluation examples selection. (3) Accuracy-based evaluation. The workflow of $D^2$Controller is illustrated in Figure 5.

### 4.1 IN-CONTEXT EXAMPLES SAMPLING

For each $k$-shot setting, we sample $N_s$ groups of in-context examples to evaluate, where $N_s$ is the number of in-context example groups. Each group of in-context examples is denoted as:

$$\mathcal{E}_i^k = \{(x_{ij}, y_{ij})|j = 1, \cdots, k|\mathcal{C}|\}, i = 1, \cdots, N_s, \qquad (3)$$

where $k$ denotes the $k$-shot setting. All in-context examples are removed from training set $\mathcal{D}$ and the remaining ones formulate the candidate set for evaluation examples, denoted as $\mathcal{D}'$.

### 4.2 IICSCORE-GUIDED EVALUATION EXAMPLES SELECTION

In traditional machine learning, there are two dimensions to assess the ability of a model: the *fit* ability and the *generalization* ability, which corresponds to how well a model can capture patterns

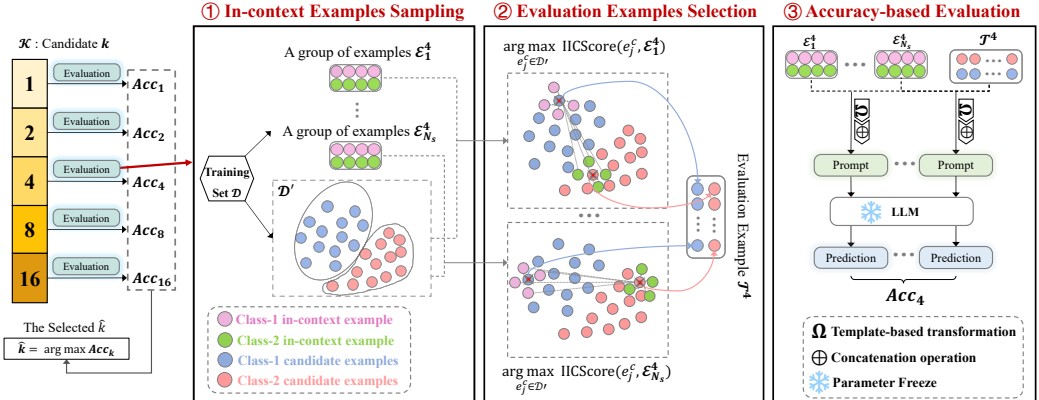

Figure 5: The whole process of the D$^2$Controller on a 2-class classification task.

in training data and deal with unseen data, respectively. Inspired by such a point of view, to comprehensively evaluate a group of in-context examples, we select similar examples (which are analogous to training-data patterns) and dissimilar examples (which are analogous to unseen data) to each class of them from the training dataset as evaluation examples. To measure similarities, we first transform each sentence $x$ to a vector representation $\boldsymbol{x}$, i.e., we input each sentence $x$ into LLMs, thereby obtaining sentence vector representations.

When searching similar examples for class-$c$ in-context examples, we expect them to be not only close to the in-context examples of class $c$, but also far from those of other classes. To this end, we propose IICScore, which considers both intra-class distance and inter-class distance, to guide our selection procedure. IICScore is defined as:

$$\text{IICScore}(e_j^c, \mathcal{E}_i^k) = -\text{KL}(\boldsymbol{x}_j^c, \bar{\boldsymbol{x}}_{\mathcal{I}_{\mathcal{E}_i^k}}^c) + \sum_{c' \in \mathcal{C}, c' \neq c} \frac{|\mathcal{D}'^{c'}|}{|\mathcal{D}'|} \text{KL}(\boldsymbol{x}_j^c, \bar{\boldsymbol{x}}_{\mathcal{I}_{\mathcal{E}_i^k}}^{c'}), \tag{4}$$

where $e_j^c = (x_j^c, y^c) \in \mathcal{D}'$ is a candidate example of class $c$, $\boldsymbol{x}_j^c$ denotes the vector representation of instance $x_j^c$, $\mathcal{I}_{\mathcal{E}_i^k}$ denotes the set of all instances in $\mathcal{E}_i^k$, $\bar{\boldsymbol{x}}_{\mathcal{I}_{\mathcal{E}_i^k}}^c$ is the average representation of class-$c$ instances in $\mathcal{I}_{\mathcal{E}_i^k}$, $\mathcal{D}'^{c'}$ means the set of class-$c'$ candidate examples, and $\text{KL}(\cdot, \cdot)$ is the KL divergence. The $\frac{|\mathcal{D}'^{c'}|}{|\mathcal{D}'|}$ is a scale factor that balances the contribution of intra-class distance and inter-class distance. Given that the $\boldsymbol{x}_j^c$ is a distribution, we choose KL divergence to measure distances. The higher the IICScore is, the more similar that candidate example $e_j^c$ is to class-$c$ in-context examples. For each group $\mathcal{E}_i^k$, the example with the highest IICScore in each class is selected as follows:

$$\tilde{e}_{\mathcal{E}_i^k}^c = \underset{e_j^c \in \mathcal{D}'}{\arg\max} \quad \text{IICScore}(e_j^c, \mathcal{E}_i^k). \tag{5}$$

In total, $|\mathcal{C}|$ similar examples are selected for each $\mathcal{E}_i^k$.

There is no need to identify dissimilar examples, however, as they have already been obtained when selecting similar examples: For any two different groups of in-context examples $\mathcal{E}_i^k$ and $\mathcal{E}_j^k$, their similar examples are different. Then the similar example $\tilde{e}_{\mathcal{E}_j^k}^c$ is naturally a dissimilar example for $\mathcal{E}_i^k$. Gathering all $N_s|\mathcal{C}|$ similar examples to form the set of evaluation examples $\mathcal{T}^k$, there are $|\mathcal{C}|$ similar examples and $(N_s - 1)|\mathcal{C}|$ less similar examples for each group of in-context examples.

## 4.3 ACCURACY-BASED EVALUATION

In the last stage, each group of in-context examples is transformed into demonstrations and each instance of evaluation examples in $\mathcal{T}^k$ is transformed into a test input. Then we iteratively concatenate demonstrations with every test input to create prompts (As shown in Equation 1). After that, the prompts are fed into LLMs to get predictions. The average prediction accuracy of $N_s$ group of

demonstrations is treated as the performance of $k$-shot setting:

$$\text{Acc}_k = \frac{1}{N_s} \sum_{i=1}^{N_s} \left( \frac{1}{|\mathcal{T}^k|} \sum_{j=1}^{|\mathcal{T}^k|} \mathbb{I}(\hat{y}_{j,\mathcal{E}_i^k} = y_j) \right), \tag{6}$$

where $\hat{y}_{j,\mathcal{E}_i^k}$ means the predicted label of $j$-th example in $\mathcal{T}^k$ using demonstrations transformed from $\mathcal{E}_i^k$ and $\mathbb{I}$ is the indicator function. After testing the performance of all feasible $k$-shot settings, we choose the one with the best performance as follows:

$$\hat{k} = \underset{k \in \mathcal{K}}{\arg\max} \quad \text{Acc}_k. \tag{7}$$

The algorithm details of the $D^2$Controller are presented in Appendix D. It is worth mentioning that our approach is model-agnostic, allowing it to be combined with LLMs of different sizes and applied to previous ICL methods.

## 5 EXPERIMENTS

### 5.1 SETUP

**Datasets**    We conduct experiments on ten text classification datasets ranging from sentiment classification to textual entailment, including SST-2 (Socher et al., 2013), SST-5 (Socher et al., 2013), DBPedia (Zhang et al., 2015), MR (Pang & Lee, 2005), CR (Hu & Liu, 2004), MPQA (Wiebe et al., 2005), Subj (Pang & Lee, 2004), AGNews (Zhang et al., 2015), RTE (Dagan et al., 2005), and CB (De Marneffe et al., 2019). More details of the datasets are provided in Appendix B.

**LLMs**    To verify the effectiveness of $D^2$Controller, we apply our method to a wide range of LLMs, including three GPT-2 models (Radford et al., 2019) (with 0.3B, 0.8B, and 1.5B parameters), two Cerebras-GPT models (Dey et al., 2023) (with 2.7B and 6.7B parameters), two OPT models (Zhang et al., 2022a) (with 13B and 30B parameters) and GPT-3 175B model (Brown et al., 2020).

**Evaluation Metric**    Following (Lu et al., 2022; Xu et al., 2023), to control the GPT-3 inference costs [2], we randomly sample 256 examples from the validation set for each dataset to evaluate the accuracy and report the average performance and standard deviation over 5 different seeds.

**Implementation Details**    For $D^2$Controller, $\mathcal{K}$ is set as $\{1, 2, 4, 8, \cdots, k_{\max}\}$ (See Appendix B for details of $k_{\max}$ of each dataset on different sizes of LLMs). We sample $N_s = 5$ groups of in-context examples for $k$-shot setting evaluation on Cerebras-GPT-2.7B model, and set $N_s$ as 25 on other sizes of LLMs, the reason of which is presented in the Section 5.4.2. We implement our method with the PyTorch framework and run experiments on 8 NVIDIA A100 GPUs.

### 5.2 BASE MODEL AND ORACLE

We consider the default $k$-shot setting in previous work (Lu et al., 2022; Xu et al., 2023) as our base model, which is: the 4-shot settting for most of the datasets except the 1-shot setting for the DBpedia dataset and the 2-shot setting for the AGNews dataset. In addition, we also provide an *Oracle* to show the upper bound of performance, that is, for each dataset, we iterate all feasible $k$-shot settings on 256 examples (mentioned in Evaluation Metric) and record the highest achievable performance.

### 5.3 MAIN RESULTS

The main experiment results are shown in Table 1, from which we have following findings:

**$D^2$Controller is effective for selecting suitable $k$-shot setting for each dataset and is compatible with different LLMs.**    In comparison to the base model, $D^2$Controller achieves 5.4% relative improvements on average across ten datasets, which validates the rationality of dynamically selecting

---

[2]It requires the usage of a monetary paid-for API.

Table 1: Main results of our methods on eight sizes of LLMs across ten datasets. We report the average performance and standard deviation over 5 different seeds for each dataset. The last column represents the average result across the ten datasets. **AVG** is short for Average.

| | | SST-2 | SST-5 | DBPedia | MR | CR | MPQA | Subj | AGNews | RTE | CB | AVG |
|---|---|---|---|---|---|---|---|---|---|---|---|---|
| GPT-2 0.3B | Default | $58.1_{13.1}$ | $24.1_{7.4}$ | $60.6_{7.2}$ | $54.2_{10.6}$ | $50.6_{0.4}$ | $59.6_{15.8}$ | $53.4_{5.3}$ | $48.7_{8.5}$ | $51.3_{1.7}$ | $48.6_{6.4}$ | 50.9 |
| | D²Controller | $74.1_{9.3}$ | $31.6_{8.6}$ | $60.6_{7.2}$ | $53.8_{7.0}$ | $67.7_{11.4}$ | $57.1_{9.7}$ | $53.8_{4.2}$ | $48.7_{8.5}$ | $48.7_{2.9}$ | $48.6_{6.4}$ | **54.5** |
| | Oracle | $74.1_{9.3}$ | $31.6_{8.6}$ | $60.6_{7.2}$ | $56.0_{9.9}$ | $67.7_{11.4}$ | $64.5_{16.0}$ | $58.6_{12.8}$ | $49.4_{18.4}$ | $51.3_{1.7}$ | $50.0_{9.2}$ | 56.4 |
| GPT-2 0.8B | Default | $71.8_{12.1}$ | $37.8_{6.8}$ | $63.4_{6.0}$ | $71.1_{15.6}$ | $80.5_{11.4}$ | $65.8_{11.3}$ | $59.9_{12.2}$ | $65.6_{17.2}$ | $53.1_{3.4}$ | $37.1_{14.5}$ | 60.6 |
| | D²Controller | $65.9_{15.2}$ | $37.5_{5.1}$ | $63.4_{6.0}$ | $71.1_{15.6}$ | $80.5_{11.4}$ | $70.5_{5.2}$ | $69.4_{12.4}$ | $65.6_{17.2}$ | $53.1_{3.4}$ | $47.5_{3.2}$ | **62.4** |
| | Oracle | $71.8_{12.1}$ | $39.6_{5.1}$ | $63.4_{6.0}$ | $71.1_{15.6}$ | $80.5_{11.4}$ | $74.5_{8.8}$ | $69.4_{12.4}$ | $65.6_{17.2}$ | $53.8_{4.4}$ | $49.3_{3.7}$ | 63.9 |
| GPT-2 1.5B | Default | $70.3_{6.6}$ | $35.4_{8.4}$ | $82.0_{2.0}$ | $52.0_{3.8}$ | $52.0_{3.2}$ | $66.7_{8.2}$ | $57.3_{10.5}$ | $78.2_{6.7}$ | $53.1_{1.7}$ | $52.9_{6.3}$ | 60.0 |
| | D²Controller | $81.3_{5.4}$ | $35.4_{8.4}$ | $82.0_{2.0}$ | $72.2_{13.9}$ | $66.2_{16.7}$ | $83.9_{1.5}$ | $64.1_{11.3}$ | $78.2_{6.7}$ | $53.1_{2.9}$ | $52.9_{6.3}$ | **67.0** |
| | Oracle | $81.3_{5.4}$ | $40.6_{5.4}$ | $82.0_{2.0}$ | $72.2_{13.9}$ | $66.2_{16.7}$ | $83.9_{1.5}$ | $64.1_{11.3}$ | $81.3_{7.5}$ | $53.1_{2.9}$ | $57.9_{9.8}$ | 68.2 |
| Cerebras-GPT 2.7B | Default | $65.5_{13.8}$ | $28.4_{4.3}$ | $81.8_{1.4}$ | $65.1_{11.2}$ | $85.8_{4.2}$ | $64.2_{11.6}$ | $69.3_{14.4}$ | $69.5_{3.2}$ | $48.1_{1.1}$ | $52.5_{9.5}$ | 63.0 |
| | D²Controller | $77.3_{7.7}$ | $34.3_{4.8}$ | $81.8_{1.4}$ | $76.0_{7.7}$ | $87.4_{1.5}$ | $81.6_{2.1}$ | $74.2_{7.6}$ | $77.3_{4.1}$ | $48.0_{1.1}$ | $54.6_{2.7}$ | **69.3** |
| | Oracle | $80.7_{9.1}$ | $34.3_{4.8}$ | $81.8_{1.4}$ | $76.0_{7.7}$ | $87.4_{1.5}$ | $82.9_{3.0}$ | $74.2_{7.6}$ | $77.3_{4.1}$ | $49.6_{2.3}$ | $55.7_{5.0}$ | 70.0 |
| Cerebras-GPT 6.7B | Default | $83.4_{8.5}$ | $38.3_{1.8}$ | $87.0_{2.4}$ | $88.0_{1.1}$ | $89.0_{3.1}$ | $75.2_{10.3}$ | $72.0_{14.5}$ | $79.2_{2.4}$ | $52.3_{2.3}$ | $52.5_{8.0}$ | 71.7 |
| | D²Controller | $82.0_{11.3}$ | $39.5_{3.7}$ | $87.0_{2.4}$ | $86.8_{1.9}$ | $90.5_{0.9}$ | $83.8_{3.3}$ | $79.2_{12.5}$ | $80.2_{1.5}$ | $52.8_{2.5}$ | $57.9_{7.2}$ | **74.0** |
| | Oracle | $88.6_{2.7}$ | $43.6_{1.6}$ | $87.0_{2.4}$ | $88.0_{1.1}$ | $90.6_{2.8}$ | $83.8_{3.3}$ | $79.2_{12.5}$ | $80.2_{1.5}$ | $53.4_{1.7}$ | $57.9_{3.0}$ | 75.2 |
| OPT 13B | Default | $81.2_{6.7}$ | $43.3_{4.6}$ | $92.3_{2.1}$ | $87.8_{2.7}$ | $91.4_{3.3}$ | $75.0_{6.7}$ | $79.1_{12.7}$ | $81.9_{2.9}$ | $54.4_{4.2}$ | $58.9_{8.1}$ | 74.5 |
| | D²Controller | $90.2_{5.8}$ | $43.3_{4.6}$ | $92.3_{2.1}$ | $87.8_{2.7}$ | $91.3_{2.1}$ | $72.0_{9.4}$ | $91.6_{2.0}$ | $82.6_{1.5}$ | $55.8_{3.1}$ | $58.9_{8.1}$ | **76.6** |
| | Oracle | $90.9_{3.7}$ | $48.0_{2.8}$ | $92.3_{2.1}$ | $91.8_{0.6}$ | $93.3_{1.2}$ | $78.6_{7.3}$ | $91.6_{2.0}$ | $82.6_{1.5}$ | $55.8_{3.1}$ | $73.2_{12.4}$ | 79.8 |
| OPT 30B | Default | $92.3_{1.3}$ | $40.9_{1.8}$ | $91.7_{3.7}$ | $91.8_{2.1}$ | $87.3_{3.3}$ | $78.8_{6.2}$ | $76.1_{4.9}$ | $78.7_{3.6}$ | $63.0_{3.1}$ | $60.0_{8.2}$ | 76.1 |
| | D²Controller | $92.3_{1.3}$ | $42.0_{2.8}$ | $91.7_{3.7}$ | $93.4_{1.1}$ | $87.3_{3.8}$ | $85.7_{3.8}$ | $83.4_{8.6}$ | $76.7_{4.5}$ | $61.6_{2.8}$ | $60.0_{8.2}$ | **77.4** |
| | Oracle | $92.8_{1.6}$ | $45.2_{3.1}$ | $91.7_{3.7}$ | $93.4_{1.1}$ | $87.7_{3.9}$ | $85.7_{3.8}$ | $83.4_{8.6}$ | $78.7_{3.6}$ | $63.0_{3.1}$ | $60.0_{8.2}$ | 78.1 |
| GPT-3 175B | Default | $94.0_{1.4}$ | $47.7_{0.6}$ | $90.2_{2.8}$ | $94.1_{0.6}$ | $91.4_{0.0}$ | $84.4_{0.6}$ | $71.1_{2.2}$ | $86.9_{1.4}$ | $60.4_{5.3}$ | $70.5_{13.9}$ | 79.1 |
| | D²Controller | $94.0_{1.4}$ | $48.4_{0.6}$ | $90.2_{2.8}$ | $95.5_{0.8}$ | $93.0_{2.3}$ | $84.4_{0.6}$ | $87.3_{4.7}$ | $86.9_{1.4}$ | $66.6_{3.0}$ | $73.2_{2.5}$ | **82.0** |
| | Oracle | $94.1_{0.0}$ | $48.4_{0.6}$ | $90.2_{2.8}$ | $95.5_{0.3}$ | $93.6_{2.8}$ | $86.5_{2.5}$ | $87.3_{4.7}$ | $86.9_{1.4}$ | $69.7_{1.4}$ | $73.2_{2.5}$ | 82.6 |

the number of demonstrations[3]. It is worth mentioning that, in contrast to other LLMs, D²Controller at most obtains 7.0% and 6.3% improvements in accuracy for GPT-2-1.5B and Cerebras-GPT-2.7B on ten datasets. These results reveal that our method has good compatibility. Some LLMs exhibit a minor decline in performance on the MPQA, SST-2, and MR datasets. One possible reason is that these datasets have relatively shorter average demonstration lengths (shown in Table 6), and they contain fewer crucial features related to classification. Therefore, selecting an appropriate demonstration number for these datasets may be more challenging.

**D²Controller achieves near-optimal results at a lower cost.** In most LLMs, our approach achieves performance levels close to that of the Oracle, aligning with our original research intent. While the Oracle represents the upper bound of performance, it is unfeasible in practice to iterate through all $k$-shot settings on large-scale examples to attain such performance, mainly due to the extensive resource and time demands. In contrast, our method achieves good performance with a small number of evaluation examples and effectively controls inference costs. Our approach underscores the practical feasibility of striking a balance between performance and resource consumption, which is a crucial aspect for a wide range of real-world applications.

## 5.4 ANALYSIS AND DISCUSSION

In this section, we conduct a series of analysis experiments related to D²Controller. It should be noted that the results we report are the average performance of ten datasets.

### 5.4.1 D²CONTROLLER IS BENEFICIAL TO OTHER ICL METHODS

Here we extend our method to some representative ICL methods, *i.e.*, applying the number of demonstrations decided by D²Controller to other ICL methods. These methods include a *Demonstration Selection* method **KATE** (Liu et al., 2022b), a *Demonstration Order* method **GlobalE** (Lu et al., 2022), and two *Calibration-based* method **Contextual Calibration** (Zhao et al., 2021) and the $k$**NN Prompting** (Xu et al., 2023). The results are shown in Table 2.

As we can see, incorporating D²Controller into other ICL methods can obtain competitive performance. To be specific, compared to KATE using the default $k$-shot settings (As mentioned in Section 5.2), KATE + D²Controller at most obtains 3.1% improvements in terms of accuracy. Similarly,

---

[3]The values of $k$ chosen by the D²Controller and Oracle are provided in Appendix E.

Table 2: The result of extending D$^2$Controller to other ICL models.

|  | GPT-2 0.3B | GPT-2 0.8B | GPT-2 1.5B | Cerebras-GPT 2.7B | Cerebras-GPT 6.7B | GPT-3 175B |
|---|---|---|---|---|---|---|
| KATE | 66.7 | 69.4 | 67.7 | 71.6 | 77.6 | 82.2 |
| + D$^2$Controller | 68.8 | 70.5 | 69.4 | 74.7 | 77.9 | 82.6 |
| GlobalE | 59.5 | 67.7 | 69.8 | - | - | - |
| + D$^2$Controller | 61.5 | 68.7 | 71.6 | - | - | - |
| Contextual Calibration | 59.5 | 64.2 | 63.9 | 67.2 | 72.5 | 78.9 |
| + D$^2$Controller | 60.8 | 66.6 | 65.4 | 68.7 | 73.5 | 80.1 |
| kNN Prompting | 74.8 | 76.0 | 77.3 | 77.8 | 79.0 | - |
| + D$^2$Controllern | 75.8 | 77.1 | 78.2 | 78.1 | 79.7 | - |

GlobalE + D$^2$Controller improves the accuracy by up to 2.0% compared to GlobalE. For Contextual Calibration and $k$NN Prompting, when combined with D$^2$Controller, the accuracy is improved by up to 2.4% and 1.1% respectively. For the GPT-3 model, integrating Contextual Calibration with D$^2$Controller enhances accuracy by 1.2%. The improvements of these extending methods further confirm the necessity to dynamically decide $k$-shot settings instead of using the default setting as well as indicate that the D$^2$Controller has excellent generalization capabilities. Moreover, the improvements in KATE + D$^2$Controller and GlobalE + D$^2$Controller prove that the number of demonstrations is a key factor in ICL performance along with the selection and ordering of demonstrations.

### 5.4.2 THE IMPACT OF THE NUMBER OF IN-CONTEXT EXAMPLE GROUPS $N_s$

To investigate the effect of the number of in-context example groups $N_s$ on D$^2$Controller, we vary the value of $N_s$ in the range of [5, 30] with a step size of 5. Figure 6 shows the average performance of D$^2$Controller with different $N_s$ on ten datasets. Actually, the majority of LLMs can achieve good results at $N_s = 5$, and their performance remains stable as the number of in-context example groups increases. For the other LLMs, their performance has an initial upward trend and then flattens out. These observations indicate that D$^2$Controller can select near-optimal $k$-shot settings depending on a small number of in-context example groups. Finally, according to the trend of the curve, we set $N_s$ to 5 in the Cerebras-GPT-2.7B model and set $N_s$ as 25 in other sizes of LLMs.

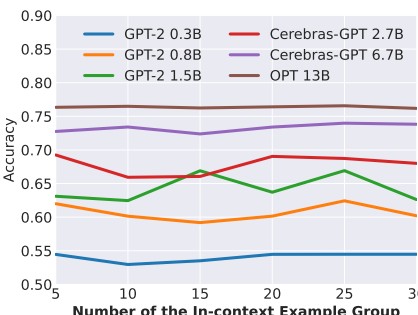

Figure 6: The impact of the number of in-context example groups $N_s$ on D$^2$Controller.

### 5.4.3 THE EFFECTIVENESS OF IICSCORE

In D$^2$Controller, we use IICScore to select evaluation examples. Here, we also explore other ways to select evaluation examples. As shown in Table 3, **Random** denotes randomly selecting the same number of examples as that of IICScore. **D$^2$Controller-ED** and **D$^2$Controller-Cos** indicate replacing KL divergence in Equation 4 with Euclidean distance and negative cosine similarity, respectively. It is clear that D$^2$Controller outperforms Random in every LLM, which suggests that the evaluation examples selected by D$^2$Controller are more representative than those of Random to properly reflect the performance of each $k$-shot setting. Comparing D$^2$Controller with the two variants, we can find that both of the variants perform worse than D$^2$Controller on most of the LLMs (except for GPT-2-0.3B), which verifies the superiority of using KL divergence as the distance metric.

### 5.4.4 DYNAMIC $k$ v.s. MAXIMUM $k$

We also compare dynamically selecting the $k$-shot setting (*i.e.*, D$^2$Controller) with using the maximum number of demonstrations (*i.e.*, $k_{\max}$-shot setting). As shown in Table 4, we observe that our method achieves more competitive results, which agree with our motivation mentioned in Section 3. Specifically, in contrast to the $k_{\max}$-shot setting, our approach achieves a 2.6% relative improvement across six different sizes of LLMs on ten datasets, indicating that adopting the $k_{\max}$-shot setting for each dataset is not appropriate. It is crucial to mention that the performance of D$^2$Controller is improved by up to 3.7% and 3.3% on the GPT-2-0.8B model and the Cerebras-GPT-2.7B model

Table 3: The results of using three other ways to select evaluation examples.

| | GPT-2 0.3B | GPT-2 0.8B | GPT-2 1.5B | Cerebras-GPT 2.7B | Cerebras-GPT 6.7B | GPT-3 175B |
|---|---|---|---|---|---|---|
| Random | 54.1 | 59.2 | 63.5 | 68.0 | 72.9 | 81.3 |
| D$^2$Controller-ED | 54.4 | 59.2 | 64.0 | 67.1 | 72.6 | 79.1 |
| D$^2$Controller-Cos | **54.9** | 59.3 | 62.2 | 68.3 | 72.4 | 80.4 |
| D$^2$Controller | 54.5 | **62.4** | **66.9** | **69.3** | **74.0** | **82.0** |

Table 4: The results of D$^2$Controller and using the maximum number of demonstrations (*i.e.*, $k_{\max}$-shot setting) for each dataset.

| | GPT-2 0.3B | GPT-2 0.8B | GPT-2 1.5B | Cerebras-GPT 2.7B | Cerebras-GPT 6.7B | GPT-3 175B |
|---|---|---|---|---|---|---|
| $k_{\max}$-shot setting | 54.1 | 58.7 | 66.0 | 65.4 | 73.0 | 81.4 |
| D$^2$Controller | 54.5 | 62.4 | 67.0 | 68.7 | 74.0 | 82.0 |

compared to other LLMs. These results further highlight the superiority of dynamic demonstration selection. In addition, our approach achieves better performance with fewer demonstrations compared to utilizing the maximum number of demonstrations. This underscores that D$^2$Controller economizes both time and space during the inference of LLMs from another perspective.

## 6 RELATED WORK

With the increase in both model size and training corpus size (Devlin et al., 2019; Radford et al., 2019; Brown et al., 2020; Chowdhery et al., 2022), large language models (LLMs) show a capacity for In-Context Learning (ICL). Given that ICL is sensitive to the selection and the order of the demonstrations (Liu et al., 2022a; Rubin et al., 2022; Zhang et al., 2022b; Lu et al., 2022; Wang et al., 2023; Wu et al., 2022; Li et al., 2023; Li & Qiu, 2023; Li et al., 2023), their works can be roughly divided into two categories:

(1) *Demonstration Selection*. Liu et al. (2022a) propose to retrieve in-context examples that are semantically similar to a test example to formulate its corresponding prompt. Rubin et al. (2022) first label training examples as positive or negative, and then train an efficient dense retriever using this data, which is used to retrieve training examples as prompts at test time. Zhang et al. (2022b) formulate the problem as a sequential decision problem, and propose a reinforcement learning algorithm for identifying generalizable policies to select demonstrations. Li & Qiu (2023) propose to find supporting examples for ICL. Specifically, they design a two-stage method to filter and search demonstrations from training data.

(2) *Demonstration Ordering*. Lu et al. (2022) study order sensitivity for ICL and propose a simple, generation-based probing method to identify performant prompts. Wu et al. (2022) propose the self-adaption mechanism to help each input find a demonstration organization (i.e., selection and permutation) that can derive the correct output, thus maximizing performance.

However, there are few studies related to the impact of the number of demonstrations within a limited input length on ICL performance. The closest work to ours is Xu et al. (2023), which proposes a method that utilizes an unlimited number of training examples for model calibration, while our research focuses on how to select an appropriate number of demonstrations for each dataset when the input length is restricted. Therefore, the two methods have different starting points.

## 7 CONCLUSION

In this paper, we conduct an in-depth analysis of the impact of the number of demonstrations on ICL performance within a limited input length of LLM. Surprisingly, we discover that the number of demonstrations does not always exhibit a positive correlation with model performance. Based on these analyses, we propose a method named D$^2$Controller, which can improve the ICL performance by dynamically adjusting the number of demonstrations. The experimental results show our method achieves an average of 5.4% relative improvement across ten datasets on eight different sizes of LLMs. Further analysis verifies the effectiveness of our method.

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

## A    DETAIL FOR DEMONSTRATION AND LABEL SPACE

As depicted in Table 5, we provide detailed information on the Demonstration, mapping token space, and label space for different tasks.

Table 5: Demonstration, mapping token space, and label space for different tasks.

| Dataset | Demonstration | Mapping Token Space $\mathcal{V}$ | Label Space $\mathcal{Y}$ |
|---|---|---|---|
| SST-2 | Review: the greatest musicians. Sentiment: Positive | positive/negative | positive/negative |
| SST-5 | Review: it 's a very valuable film ... Sentiment: great | terrible/bad/okay /good/great | very positive/positive /neutral/negative /very negative |
| DBPedia | input: Monte Vermenone is a mountain of Marche Italy. type: nature | company/school/artist/ athlete/politics/book/ building/nature/village/ animal/plant/album/ film/transportation | company/school/artist/ athlete/politics/book/ building/nature/village/ animal/plant/album/ film/transportation |
| MR | Review: a dreary movie . Sentiment: negative | positive/negative | positive/negative |
| CR | Review: i am bored with the silver look . Sentiment: negative | positive/negative | positive/negative |
| MPQA | Review: is also the most risky Sentiment: negative | positive/negative | positive/negative |
| Subj | Input: presents a most persuasive vision of hell on earth . Type: subjective | subjective/objective | subjective/objective |
| AGNews | input: Historic Turkey-EU deal welcomed. The European Union's decision to hold entry talks with Turkey receives a widespread welcome. type: world | world/sports/business /technology | world/sports/business /technology |
| RTE | premise: Oil prices fall back as Yukos oil threat lifted hypothesis: Oil prices rise. prediction: not_entailment | true/false | entailment/not_entailment |
| CB | premise: "Clever". Klug means "clever". Would you say that Abie was clever? hypothesis: Abie was clever prediction: neutral | true/false/neither | entailment/contradiction/ neutral |

## B    DETAIL FOR DATASETS AND MAX SHOTS

As shown in Table 6, we present detailed information for ten datasets. Besides, as we mentioned in section 2.1, for each dataset, the input prompt $P$ consists of different numbers of demonstrations and a test instance. The maximum shot number, *i.e.*, $k_{\max}$ is calculated as follows:

$$\text{Upper}_{bound} = \frac{\text{Max}_{input} - \text{Max}_{test}}{\text{Avg}_{template} * \text{Numbers}_{classes}}, \qquad (8)$$

$$k_{\max} = \max 2^i \leq \text{Upper}_{bound}, \quad i = 0, 1, 2, \cdots \qquad (9)$$

where $\text{Upper}_{bound}$ is the Upper-bound of shots that can be accommodated by GPT-2, Cerebras-GPT, OPT or GPT-3, $\text{Max}_{input}$ indicates the maximum input length of different sizes of LLMs, i.e., GPT-2 (1024 tokens), Cerebras-GPT-2.7B (2048 tokens), Cerebras-GPT-6.7B (2048 tokens), OPT-13B (2048 tokens), OPT-30B (2048 tokens), GPT-3 175B (2048 tokens), $\text{Max}_{test}$ denotes the max length of test input, $\text{Avg}_{template}$ means the average length of each demonstration, and $\text{Numbers}_{classes}$ indicates the numbers of classes for each task, *i.e.*, $|\mathcal{C}|$. To narrow down the search scope, we set the value range of Max Shots to $\{1, 2, 4, 8, 16, 32, 64, \cdots\}$. Thus, for each dataset, the max shots we choose should be below the upper bound and closest to it. For example, the Upper-bound (1024 tokens) of the SST-2 dataset is 25, so the max shot we need to select is 16; the Upper-bound (1024 tokens) of the MPQA dataset is 48, so the max shot we need to select is 32. It should be noted that while the Upper-bound (1024 tokens) of the CB dataset is 2, for a fair comparison with other

Table 6: Statistics of evaluation datasets, the average length of each demonstration, and the max length of test input are calculated based on sentence-piece length.

| Dataset | Number of Classes | Avg. Length of Demonstration | Max Length of Test Input | Upper-bound (1024 tokens) | Max Shots (1024 tokens) | Upper-bound (2048 tokens) | Max Shots (2048 tokens) |
|---|---|---|---|---|---|---|---|
| SST-2 (Socher et al., 2013) | 2 | 19.1 | 55 | 25 | 16 | 52 | 32 |
| SST-5 (Socher et al., 2013) | 5 | 29.7 | 60 | 6 | 4 | 13 | 8 |
| DBPedia (Zhang et al., 2015) | 14 | 71.6 | 161 | 1 | 1 | 1 | 1 |
| MR (Pang & Lee, 2005) | 2 | 32.7 | 66 | 14 | 8 | 30 | 16 |
| CR (Hu & Liu, 2004) | 2 | 29.0 | 99 | 15 | 8 | 33 | 32 |
| MPQA (Wiebe et al., 2005) | 2 | 10.4 | 19 | 48 | 32 | 97 | 64 |
| Subj (Pang & Lee, 2004) | 2 | 34.9 | 91 | 13 | 8 | 28 | 16 |
| AGNews (Zhang et al., 2015) | 4 | 59.5 | 167 | 3 | 2 | 7 | 4 |
| RTE (Dagan et al., 2005) | 2 | 79.7 | 256 | 4 | 4 | 11 | 8 |
| CB (De Marneffe et al., 2019) | 3 | 90.8 | 278 | 2 | 4 | 6 | 4 |

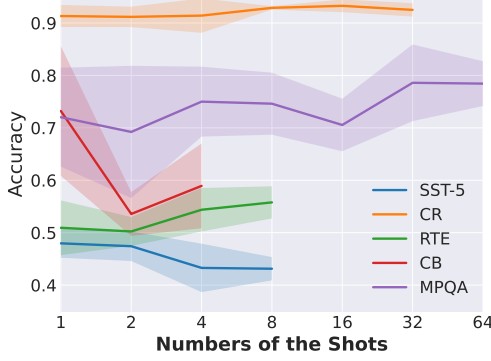

Figure 7: Effect of the number of demonstrations on OPT-13B across five datasets.

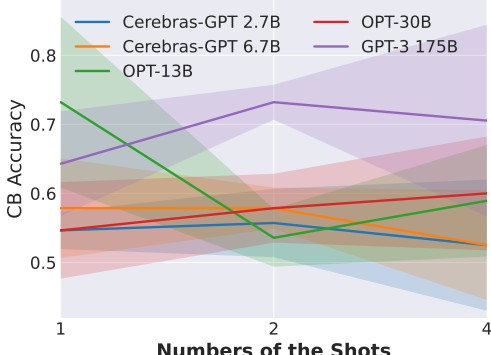

Figure 8: The accuracy of five different sizes of LLMs on the CB dataset.

methods, we set the max shot to 4. This decision was made because previous methods used 4-shots for the CB dataset (Lu et al., 2022).

## C  ADDITIONAL RESULTS

Here, we present more results to support our arguments. Among them, Figure 7 shows the performance curves of five datasets on the OPT-13B model. Figure 8 shows performance curves of CB dataset on five different sizes of LLMs.

**Increasing the number of demonstrations does not necessarily improve the model performance.** In Figure 7, when changing from 1-shot setting to $k_{max}$-shot setting, we can observe that the accuracy of the OPT-13B model improves on the RTE and MPQA datasets while declines on the SST5 and CB datasets. Besides, as shown in Figure 8, when changing from 1-shot setting to 4-shot setting, the accuracy of the CB dataset initially declines and then increases on the OPT-13B model, while it first rises and then goes down on the GPT-3-175B model. These observations suggests that the inclusion of more demonstrations does not guarantee improved performance.

**The optimal $k$-shot setting differs depending on specific datasets and models.** From Figure 8, we can find that the optimal $k$-shot settings for the same dataset on different models can be different: 1-shot setting for the OPT-13B model, 2-shot setting for the Cerebras-GPT 2.7B, Cerebras-GPT 6.7B and GPT-3 175B models, 4-shot setting for the OPT-30B model. Likewise, from Figure 7, we can tell that the optimal $k$-shot settings for the same model on different datasets also can be different: 1-shot setting for the SST5 and CB datasets, 8-shot setting for the RTE dataset, 16-shot setting for the CR dataset, 32-shot setting for the MPQA dataset. These observations suggests that the optimal number of demonstrations may differ depending on the specific dataset and model.

We speculate that adding a demonstration to a prompt will have two effects: (1) Providing more information to the prompt, resulting in improvement in performance. (2) Causing the distribution of the prompt to become more different from the pre-training corpus of LLMs, leading to difficulty

in understanding the prompt and reducing performance. When more demonstrations are added, the direction of the change in performance depends on which effect is more influential. For different datasets and LLMs, when adding more demonstrations, the strengths of Effect (1) and Effect (2) are different, leading to the variation observed in pilot experiments and also causing the difference in the optimal $k$.

## D  ALGORITHM DETAILS

The details of Dynamic Demonstrations Controller are presented in Algorithm 1.

---

**Algorithm 1:** Dynamic Demonstrations Controller.

**Input:** The training set: $\mathcal{D}$; The number of in-context example groups: $N_s$; The feasible $k$ set: $\mathcal{K}$; The set of Classes: $\mathcal{C}$; The LLM: $\theta$.

**Output:** The selected $k$: $\hat{k}$.

1 **for** $k$ *in* $\mathcal{K}$ **do**
2      Sampling $N_s$ groups of in-context examples and remove them from $\mathcal{D}$. The rest is $\mathcal{D}'$.
     // Initializing the set of evaluation examples.
3      $\mathcal{T}^k \leftarrow \emptyset$
4      **for** $i$ *in* $1, 2, \cdots, N_s$ **do**
5          **for** $c$ *in* $\mathcal{C}$ **do**
             // Computing the IICScore for each candidate example in $\mathcal{D}'$.
6              $\tilde{e}^c_{\mathcal{E}^k_i} \leftarrow \underset{e^c_j \in \mathcal{D}'}{\arg\max} \ \ \text{IICScore}(e^c_j, \mathcal{E}^k_i)$
7              $\mathcal{T}^k \leftarrow \mathcal{T}^k \cup \tilde{e}^c_{\mathcal{E}^k_i}$
8          **end**
9      **end**
10      Acc $\leftarrow 0$
11      **for** $i$ *in* $1, 2, \cdots, N_s$ **do**
12          Acc $\leftarrow$ Acc $+ \frac{1}{|\mathcal{T}^k|} \sum_{j=1}^{|\mathcal{T}^k|} \mathbb{I}(\hat{y}_{j,\mathcal{E}^k_i} = y_j)$
13      **end**
14      $\text{Acc}_k \leftarrow \frac{1}{N_s} \text{Acc}$
15 **end**
16 $\hat{k} \leftarrow \underset{k \in \mathcal{K}}{\arg\max} \ \ \text{Acc}_k$
17 **return** $\hat{k}$

---

## E    THE VALUE OF $k$

In Table 7, we show the values of $k$ chosen by the D$^2$Controller and *Oracle*.

Table 7: The values of $k$ chosen by the D$^2$Controller and *Oracle*.

| | | SST-2 | SST-5 | DBPedia | MR | CR | MPQA | Subj | AGNews | RTE | CB |
|---|---|---|---|---|---|---|---|---|---|---|---|
| GPT-2 0.3B | Default | 4 | 4 | 1 | 4 | 4 | 4 | 4 | 2 | 4 | 4 |
| | D$^2$Controller | 16 | 1 | 1 | 8 | 1 | 32 | 2 | 2 | 2 | 4 |
| | Oracle | 16 | 1 | 1 | 1 | 1 | 16 | 8 | 1 | 4 | 2 |
| GPT-2 0.8B | Default | 4 | 4 | 1 | 4 | 4 | 4 | 4 | 2 | 4 | 4 |
| | D$^2$Controller | 16 | 2 | 1 | 4 | 4 | 32 | 8 | 2 | 4 | 2 |
| | Oracle | 4 | 1 | 1 | 4 | 4 | 16 | 8 | 2 | 2 | 1 |
| GPT-2 1.5B | Default | 4 | 4 | 1 | 4 | 4 | 4 | 4 | 2 | 4 | 4 |
| | D$^2$Controller | 16 | 4 | 1 | 8 | 8 | 16 | 8 | 2 | 2 | 4 |
| | Oracle | 16 | 1 | 1 | 8 | 8 | 16 | 8 | 1 | 2 | 2 |
| Cerebras-GPT 2.7B | Default | 4 | 4 | 1 | 4 | 4 | 4 | 4 | 2 | 4 | 4 |
| | D$^2$Controller | 32 | 8 | 1 | 16 | 1 | 32 | 16 | 1 | 4 | 1 |
| | Oracle | 8 | 8 | 1 | 16 | 1 | 64 | 16 | 1 | 2 | 2 |
| Cerebras-GPT 6.7B | Default | 4 | 4 | 1 | 4 | 4 | 4 | 4 | 2 | 4 | 4 |
| | D$^2$Controller | 32 | 2 | 1 | 8 | 32 | 64 | 16 | 4 | 8 | 1 |
| | Oracle | 1 | 1 | 1 | 4 | 16 | 64 | 16 | 4 | 2 | 2 |
| OPT 13B | Default | 4 | 4 | 1 | 4 | 4 | 4 | 4 | 2 | 4 | 4 |
| | D$^2$Controller | 16 | 4 | 1 | 4 | 1 | 1 | 16 | 4 | 8 | 4 |
| | Oracle | 1 | 1 | 1 | 1 | 16 | 32 | 16 | 4 | 8 | 1 |
| OPT 30B | Default | 4 | 4 | 1 | 4 | 4 | 4 | 4 | 2 | 4 | 4 |
| | D$^2$Controller | 4 | 8 | 1 | 16 | 2 | 64 | 16 | 4 | 8 | 4 |
| | Oracle | 2 | 1 | 1 | 16 | 16 | 64 | 16 | 2 | 4 | 4 |
| GPT-3 175B | Default | 4 | 4 | 1 | 4 | 4 | 4 | 4 | 2 | 4 | 4 |
| | D$^2$Controller | 4 | 8 | 1 | 16 | 1 | 4 | 16 | 2 | 2 | 2 |
| | Oracle | 2 | 8 | 1 | 8 | 2 | 32 | 16 | 2 | 8 | 2 |

## F    PILOT EXPERIMENTS ON GPT-4

Similar to Section 3, we conduct pilot experiments with the GPT-4 model on five text classification datasets. Due to budgetary constraints, for each dataset, we use five different seeds to test the model's performance in the 1-shot setting, the default setting (4-shot), and $k_{max}$-shot setting. Note that the maximum input length of the GPT-4 model we use is 8192 tokens, so the maximum shot number for SST-5, CR, MPQA, RTE, and CB is 32, 128, 256, 32, and 16. The results are shown in Table 8.

From the perspective of a general trend, when the input increases from a 1-shot setting to $k_{max}$- shot setting, the accuracy improves on the CR, MPQA, and RTE datasets while declines on the SST-5 and CB datasets. Moreover, the RTE dataset achieves the best performance in the default setting, rather than $k_{max}$- shot setting. Thus, increasing the number of demonstrations in stronger LLM like GPT-4 does not necessarily improve performance.

## G    D$^2$CONTROLLER *v.s.* VALIDATION SETS

we randomly sample more examples as a baseline to select $k$. Specifically, we construct three different sizes of validation sets (100, 200, and 300) to select $k$. The results are shown in Table 9 (note that the results we report are the average performance of ten datasets).

Based on these results, we can observe that using more examples does not lead to the optimal choice of $k$, and almost all of the results are inferior to D$^2$Control. This further underscores the effectiveness of using IICScore to select a small number of representative examples.

Table 8: The results of using the 1-shot setting, default setting, and the $k_{\max}$-shot setting on GPT-4.

| GPT-4 | SST-5 | CR | MPQA | RTE | CB |
|---|---|---|---|---|---|
| 1-shot setting | $45.3_{4.4}$ | $83.7_{1.3}$ | $67.4_{1.0}$ | $82.7_{3.0}$ | $89.3_{1.8}$ |
| Default setting | $45.7_{5.0}$ | $92.2_{2.2}$ | $83.8_{0.3}$ | $89.1_{1.4}$ | $83.9_{2.5}$ |
| $k_{\max}$-shot setting | $43.6_{0.8}$ | $95.9_{0.3}$ | $90.2_{1.1}$ | $88.7_{0.6}$ | $82.7_{1.0}$ |

Table 9: The results of using validation set sampled from the training dataset.

| | GPT-2 1.5B | Cerebras-GPT 2.7B | Cerebras-GPT 6.7B | OPT 13B |
|---|---|---|---|---|
| Default | 60.0 | 63.0 | 71.7 | 74.5 |
| Validation-100 | 64.9 | 68.3 | 72.6 | 75.8 |
| Validation-200 | 65.4 | 68.5 | 71.8 | 76.1 |
| Validation-300 | 64.9 | 68.3 | 72.6 | 76.4 |
| D$^2$Controller | **67.0** | **69.3** | **74.0** | **76.6** |

## H  USING DIFFERENT RETRIEVAL MODELS

In this section, we try another two text encoders (i.e., BERT-large and RoBERTa-large) to obtain sentence representations $\mathbf{x}$. The results are shown in Table 10.

We can observe that D$^2$Controller(BERT-large) and D$^2$Controller(RoBERTa-large) perform worse than D$^2$Controller on most of the LLMs (except for OPT 13B), which verifies the superiority of using GPT-architecture LLMs as the text encoders to measure data similarity in representation space.

## I  THE NUMBER OF TOKENS USING DIFFERENT METHODS

In this section, we report the average number of tokens used by three methods (default $k$, maximum $k$, and D$^2$Controller) to query LLM.

Based on results in Table 11, we can observe that our method uses fewer tokens to achieve better performance compared to maximum $k$. Especially on some LLMs, such as Cerebras-GPT 2.7B and OPT-13B, D$^2$Controller saves almost 30% and 50% tokens. Meanwhile, although our method uses more tokens compared to the default $k$, it achieves an average relative improvement of $5.4\%$ on ten datasets.

## J  THE RUNNING TIMES FOR D$^2$CONTROLLER

In this section, we provide running times for three different sizes of LLMs during the **Evaluation Examples Selection** and **Accuracy-based Evaluation** stages in Table 12, respectively.

## K  LIMITATIONS

The current research suffers from two limitations: (1) Due to budget constraints and insufficient GPU memory, we are unable to conduct experiments on larger-scale language models; (2) Our method does not guarantee the selection of the optimal value of $k$ for each dataset. Regarding the D$^2$Controller, some LLMs exhibit a minor decline in performance on the MPQA, SST-2, and MR datasets compared to the default setting. The reason behind this may be that these datasets have relatively shorter average demonstration lengths (shown in Table 6), leading to encoded semantic representations that contain less information. Thus, the similarities measured by IICScore based on these representations are inaccurate. In this case, selecting an appropriate demonstration number for these datasets may be more challenging. This requires future research to explore and refine techniques in order to continuously approach the optimal value of $k$.

Table 10: The results of using BERT-family models as text encoders.

|  | GPT-2 1.5B | Cerebras-GPT 2.7B | Cerebras-GPT 6.7B | OPT 13B |
|---|---|---|---|---|
| $D^2$Controller(BERT-large) | 65.8 | 66.5 | 71.8 | 76.6 |
| $D^2$Controller(RoBERTa-large) | 66.0 | 64.6 | 72.8 | **77.4** |
| $D^2$Controller | **67.0** | **69.3** | **74.0** | 76.6 |

Table 11: The number of tokens used by default $k$, maximum $k$, and $D^2$Controller

|  | GPT-2 1.5B | Cerebras-GPT 2.7B | Cerebras-GPT 6.7B | OPT 13B |
|---|---|---|---|---|
| Default $k$ | 455.49 | 516.87 | 516.87 | 516.87 |
| Maximum $k$ | 678.29 | 1345.72 | 1345.72 | 1345.72 |
| $D^2$Controller | 603.98 | 885.51 | 1187.37 | 725.89 |

Table 12: The running times for three different sizes of LLMs during the **Evaluation Examples Selection** and **Accuracy-based Evaluation** stages.

|  | SST-2 | SST-5 | MR | CR | MPQA | Subj | AGNews | RTE | CB |
|---|---|---|---|---|---|---|---|---|---|
| **GPT-2 1.5B** | | | | | | | | | |
| Evaluation Examples Selection | 1364 s | 313 s | 158 s | 31 s | 189 s | 140 s | 1900 s | 36 s | 10 s |
| Accuracy-based Evaluation | 915 s | 1978 s | 753 s | 654 s | 1112 s | 806 s | 1105 s | 904 s | 1987 s |
| **Cerebras-GPT 2.7B** | | | | | | | | | |
| Evaluation Examples Selection | 1662 s | 356 s | 183 s | 22 s | 197 s | 158 s | 2943 s | 47 s | 10 s |
| Accuracy-based Evaluation | 2360 s | 5386 s | 1946 s | 3654 s | 2778 s | 2096 s | 3242 s | 2419 s | 2694 s |
| **Cerebras-GPT 6.7B** | | | | | | | | | |
| Evaluation Examples Selection | 1685 s | 405 s | 189 s | 21 s | 188 s | 170 s | 2825 s | 45 s | 10 s |
| Accuracy-based Evaluation | 4832 s | 10725 s | 3942 s | 7076 s | 5558 s | 4223 s | 6432 s | 4773 s | 5376 s |

