# OpenReview forum: "Dynamic Demonstrations Controller for In-Context Learning"
_ICLR.cc/2024/Conference — Submitted to ICLR 2024_

### Official Review · Reviewer_NpsS · 2023-10-29

**Soundness:** 2 fair
**Presentation:** 3 good
**Contribution:** 2 fair
**Rating:** 5
**Confidence:** 4

**Summary:**

This paper proposed a dynamic demonstration controller to select the optimal number of demonstrations in the prompt. The proposed method can achieve similar performance with Oracle demonstration selection across different datasets and across different models. The proposed method can be integrated with existing prompt selection methods to achieve higher performance.

**Strengths:**

- The paper is clearly written and easy to follow.

 - The ablation study is comprehensive.

**Weaknesses:**

- The evaluation and discussion can be further improved.

    - It would be interesting to discuss what causes the observations in the pilot experiments.

     - It is important to conduct experiments comparing demonstration number selection and demonstration selection. The original Table 2 shows that the proposed method can further improve based on the demonstration selection. However, it is still unclear which one is more effective among demonstration selection and dynamic demonstration number selection.


 - The limitation of the method is not fully discussed.

**Questions:**

- In terms of the limitations, when will the method fail, and when will the method have a good performance?

 - In Table 2, can you also show the performance of the “default” method, which is randomly sampling k-shot demonstrations? And also show the performance of “D2Contoller” along? It will be helpful to understand which one is more effective among demonstration selection and demonstration number selection.

 - What may cause the observations in Pilot experiments? For instance, in Figure 2, what aspects of the datasets cause the different optimal k for different datasets?

---

> ### Author Response · Authors · 2023-11-19
> **Response to Reviewer NpsS**
>
> Thanks for your appreciation of our clearly written, easy-to-follow, and comprehensive ablation study. We answer your concerns as follows:
>
> > **Q1: It would be interesting to discuss what causes the observations in the pilot experiments. For instance, in Figure 2, what aspects of the datasets cause the different optimal k for different datasets?**
>
> **A1**: Thanks for your constructive feedback. We speculate that adding a demonstration to a prompt will have two effects: (1) Providing more information to the prompt, resulting in improvement in performance. (2) Causing the distribution of the prompt to become more different from the pre-training corpus of LLMs, leading to difficulty in understanding the prompt and reducing performance. When more demonstrations are added, the direction of the change in performance depends on which effect is more influential.  For different datasets and LLMs, when adding more demonstrations, the strengths of Effect (1) and Effect (2) are different, leading to the variation observed in pilot experiments and also causing the difference in the optimal k.
>
>
>
> > **Q2: It is still unclear which one is more effective among demonstration selection and dynamic demonstration number selection.**
>
> **A2**: Thanks for your precious suggestion. As we demonstrated in our paper, KATE (a demonstration selection method) obtains more competitive performance when adapted to these ten classification datasets than our demonstration number selection method. However, KATE uses a default number to select demonstrations and is not able to decide how many demonstrations to select is good. Our method is orthogonal to KATE while also being complementary. Thus, combining two methods can achieve better performance.
>
> > **Q3: In terms of the limitations, when will the method fail, and when will the method have good performance?**
>
> **A3**: Thanks for your valuable comment. Regarding the $D^2Controller$ method, some LLMs exhibit a minor decline in performance on the MPQA, SST-2, and MR datasets compared to the default setting. One possible reason is that these datasets have relatively shorter average demonstration lengths (shown in Table 6), leading to encoded semantic representations that contain less information. Thus, the similarities measured by IICScore based on these representations are inaccurate. In this case, selecting an appropriate demonstration number for these datasets may be more challenging. In contrast, most of the datasets with relatively longer average demonstration lengths perform well with $D^2Controller$.

---

> > ### Author Response · Authors · 2023-11-22
> > **Seeking Confirmation on Resolved Concerns and Final Rating with Gratitude**
> >
> > Dear Reviewer NpsS,
> >
> > We sincerely thank you for the valuable time and effort you have invested in reviewing our paper. As the discussion period draws to a close, with less than 24 hours remaining, we are keen to understand if the concerns you raised in the initial reviews have been effectively addressed. We are open to and appreciate any further feedback, and remain committed to making necessary improvements. We would be grateful if you find that these concerns have been resolved and could reflect this in your final rating of our paper.
> >
> > Best, The authors

---

### Official Review · Reviewer_tTZn · 2023-10-30

**Soundness:** 3 good
**Presentation:** 4 excellent
**Contribution:** 2 fair
**Rating:** 5
**Confidence:** 3

**Summary:**

This paper aims to determine the optimal number of example demonstrations for in-context learning. The authors argue that the common belief that the number of demonstrations is positively correlated with model performance does not necessarily hold true. Therefore, it is critical to decide on the optimal number of example demonstrations. In this work, the authors propose a method to select representative in-context learning examples that minimize intra-class distance and maximize inter-class distance for each group of in-context examples from the training dataset. They then use these selected examples as a validation set to adjust the number of demonstrations dynamically. The authors perform experiments on a wide range of datasets and demonstrate the effectiveness of their proposed method.

**Strengths:**

1. The authors have done an excellent job of motivating the problem and providing a thorough description of their research. The paper is well-written in a high-standard and easy to understand.

2. The authors have conducted extensive experiments to demonstrate that the length of in-context learning examples is not necessarily better. Furthermore, the experimental evaluation shows that their proposed method has promising performance.

3. Validation set selection is critical to in-context learning. Compared to other works, the authors propose a method to carefully curate a representative validation set. It is meaningful and makes sense.

**Weaknesses:**

The novelty of this paper is my main concern. The idea of minimizing intra-class distance and maximizing inter-class distance has been widely used in previous machine learning works [1][2]. Similarly, the paradigm of using a validation set to choose in-context learning examples/tune in-context learning hyperparameters has also been well-explored in previous works [3][4]. If the author can provide more content to illustrate their unique contribution, I will consider improving my score.

[1] Nadagouda, N., Xu, A., & Davenport, M. A. (2023, July). Active metric learning and classification using similarity queries. In Uncertainty in Artificial Intelligence (pp. 1478-1488). PMLR.

[2] Hoffer, E., & Ailon, N. (2015). Deep metric learning using triplet network. In Similarity-Based Pattern Recognition: Third International Workshop, SIMBAD 2015, Copenhagen, Denmark, October 12-14, 2015. Proceedings 3 (pp. 84-92). Springer International Publishing.

[3] Chang, T. Y., & Jia, R. (2023, July). Data curation alone can stabilize in-context learning. In Proceedings of the 61st Annual Meeting of the Association for Computational Linguistics (Volume 1: Long Papers) (pp. 8123-8144).

[4] Li, X., & Qiu, X. (2023). Finding supporting examples for in-context learning. arXiv preprint arXiv:2302.13539.

**Questions:**

Could the authors provide more information on the cost of “Evaluation Examples Selection” and “Accuracy-based Evaluation” stages?

---

> ### Author Response · Authors · 2023-11-19
> **Response to Reviewer tTZn**
>
> Thanks for your appreciation of our excellent job, high-standard writing, and meaningful method. We address your concerns as follows:
>
> > **Q1: About novelty and contirbutions.**
>
> **A1**: Thanks for your valuable comment. The core contribution of this paper comes from two aspects:
>
> * **Contribution 1**: We refute the prevailing belief that as the number of demonstrations increases, ICL performance continues to improve. Instead, the optimal numbers of demonstrations vary on different datasets and LLMs;
>
> * **Contribution 2**: Based on the above observation, we propose a method named $D^2Controller$ to dynamically select the number of demonstrations, which is highly scalable and can not only be applied to LLMs but is also compatible with previous ICL methods.
>
>   The core contribution of  $D^2Controller$ is a fresh perspective of selecting small-scale but comprehensive evaluation examples to provide an accurate evaluation of the demonstrations.
>
>   * **Novelty**: To the best of our knowledge, previous work [1] [2] mentioned above both randomly sample examples as a validation set. Different from them, we construct small but comprehensive evaluation examples.
>
>   * **Implementation**: To realize the idea, according to the characteristics of the classification task, we adopt the approach of  "minimizing intra-class distance and maximizing inter-class distance". Of course, the approach is replaceable, as long as it can achieve our idea.
>
>   * **Effectiveness**:  To verify the effectiveness of evaluation examples selected by our method, we compare our method with using randomly sampled validation examples (With 3 different sizes, 100, 200 and 300). The results are as follows:
>
> |                     | GPT-2 1.5B | Cerebras-GPT 2.7B | Cerebras-GPT 6.7B | OPT 13B |
> | ------------------- | ---------- | ----------------- | ----------------- | ------- |
> | **Default setting** | 60.0         | 63.0                | 71.7              | 74.5    |
> | **Validation-100**  | 64.9       | 68.3              | 72.6              | 75.8    |
> | **Validation-200**  | 65.4       | 68.5              | 71.8              | 76.1    |
> | **Validation-300**  | 64.9       | 68.3              | 72.6              | 76.4    |
> | $D^2Controller$     | **67.0**         | **69.3**              | **74.0**                | **76.6**    |
>
> Based on these results, we can observe that $D^2Controller$ uses fewer examples but achieves better results compared to random sample examples.
>
> > **Q2: Could the authors provide more information on the cost of** **“Evaluation Examples Selection” and “Accuracy-based Evaluation” stages?**
>
> **A2**: We appreciate your interest in understanding the cost of these stages. In the following, we provide running times for three different sizes of LLMs during the **Evaluation Examples Selection** and **Accuracy-based Evaluation** stages, respectively.
>
> |                               | SST2  | SST5   | MR    | CR    | MPQA  | SubJ  | AGNews | RTE   | CB    |
> | ----------------------------- | ----- | ------ | ----- | ----- | ----- | ----- | ------ | ----- | ----- |
> | **GPT-2 1.5B**                |       |        |       |       |       |       |        |       |       |
> | Evaluation Examples Selection | 1364s | 313s   | 158s  | 31s   | 189s  | 140s  | 1900s  | 36s   | 10s   |
> | Accuracy-based Evaluation     | 915s  | 1978s  | 753s  | 654s  | 1112s | 806s  | 1105s  | 904s  | 1987s |
> | **Cerebras-GPT 2.7B**         |       |        |       |       |       |       |        |       |       |
> | Evaluation Examples Selection | 1662s | 356s   | 183s  | 22s   | 197s  | 158s  | 2943s  | 47s   | 10s   |
> | Accuracy-based Evaluation     | 2360s | 5386s  | 1946s | 3654s | 2778s | 2096s | 3242s  | 2419s | 2694s |
> | **Cerebras-GPT 6.7B**         |       |        |       |       |       |       |        |       |       |
> | Evaluation Examples Selection | 1685s | 405s   | 189s  | 21s   | 188s  | 170s  | 2825s  | 45s   | 10s   |
> | Accuracy-based Evaluation     | 4832s | 10725s | 3942s | 7076s | 5558s | 4223s | 6432s  | 4773s | 5376s |
>
> **References:**
>
> [1] Chang, T. Y., & Jia, R. (2023, July). Data curation alone can stabilize in-context learning. In Proceedings of the 61st Annual Meeting of the Association for Computational Linguistics (Volume 1: Long Papers) (pp. 8123-8144).
>
> [2] Li, X., & Qiu, X. (2023). Finding supporting examples for in-context learning. arXiv preprint arXiv:2302.13539.

---

> > ### Author Response · Authors · 2023-11-22
> > **Seeking Confirmation on Resolved Concerns and Final Rating with Gratitude**
> >
> > Dear Reviewer tTZn,
> >
> > We sincerely thank you for the valuable time and effort you have invested in reviewing our paper. As the discussion period draws to a close, with less than 24 hours remaining, we are keen to understand if the concerns you raised in the initial reviews have been effectively addressed. We are open to and appreciate any further feedback, and remain committed to making necessary improvements. We would be grateful if you find that these concerns have been resolved and could reflect this in your final rating of our paper.
> >
> > Best, The authors

---

> > > ### Comment · Reviewer_tTZn · 2023-11-22
> > > **To authors**
> > >
> > > Thank you for your response. While some of my concerns have been addressed, I still maintain that the contribution of this paper is fair. Your statement, ‘we adopt the approach of minimizing intra-class distance and maximizing inter-class distance. Of course, the approach is **replaceable**, as long as it can achieve our idea,’ does not make sense to me. If you believe that the core contribution of this paper is simply to conclude that ‘the optimal numbers of demonstrations vary on different datasets and LLMs,’ then your title should not be ‘Dynamic Demonstrations Controller.’ However, if the proposed method is also a main contribution of your paper that you wish to claim, then it should be highlighted as such. As I mentioned earlier, minimizing intra-class distance and maximizing inter-class distance is an idea widely used in machine learning and cannot be regarded as your core contribution. Therefore, I will keep my score.

---

> ### Author Response · Authors · 2023-11-23
> **Thanks for your response**
>
> Thanks for your feedback. Regarding the $D^2Controller$, the core contribution is we propose a fresh perspective of selecting small but comprehensive evaluation examples to provide an accurate evaluation of the demonstrations.  To achieve comprehensiveness, we co-opt the idea of the fit ability and the generalization ability in ML and adapt it to the ICL for the first time, to the best of our knowledge. As an initial attempt to implement the idea, we propose IICScore to guide us to select examples. The reason we say IICScore is "replacable" is because there may be other approaches can also achieve this goal, which leaves room for others to explore. Thanks for your response again.

---

### Official Review · Reviewer_t5tL · 2023-10-30

**Soundness:** 3 good
**Presentation:** 3 good
**Contribution:** 2 fair
**Rating:** 8
**Confidence:** 4

**Summary:**

In this work, the authors study in-context learning and how different numbers of in-context examples affect an LLM's performance on a classification task. Specifically, the authors design and conduct some pilot experiments and report that a large number of in-context examples does not always guarantee the best model performance. Motivated by this, the authors propose the $D^2$ Controller, a method that dynamically determines an optimal number $k$ for $k$-shot ICL given a dataset.

In the $D^2$ Controller algorithm, for a given $k$ value,
- first, $N_s$ groups of in-context examples are sampled;
- second, for each group, a set of evaluation data points is selected, according to the proposed IICScore, which measures the similarity between a evaluation data point and the in-context examples in the group;
- third, the accuracy of an LLM on the selected evaluation data, using the corresponding in-context examples, is obtained;
- averaging all the above accuracy scores over the $N_s$ groups resulting an overall score for $k$;
- finally, the optimal $k$ is selected according to the setting produces the highest averaged accuracy.

In experiments, the authors include a wide range of LLMs, including open-sourced LLMs and ones that can only be accessed via online APIs. The author also include 10 classification datasets. Results suggest that their method can indeed determine a better $k$ value than default settings used in prior works.

**Strengths:**

1. Useful topic: As the authors describe, there are few work studying how the number of demonstrations impacts an LLM's performance in the ICL setting. I agree this is an important topic because empirically the study could benefit millions of LLM practitioners.

2. Neat idea: I think the method is well designed, I especially like the IICScore part, where it takes both inter- and intra-class similarity into consideration.

3. Experiments and results: The authors study their methods on a wide range of LLMs and ten datasets. The authors run five seeds and report average / standard deviation. Results show that the proposed method outperforms baselines. The authors also provide a list of ablation study to help understand their method.

**Weaknesses:**

Please see my questions and concerns below.

**Questions:**

### Questions and concerns
Q1. How does $D^2$ Controller compare to a simple baseline where $k$ is optimized as a hyperparameter using a validation set?

Q2. How does $D^2$ Controller work with `classification` tasks where options have no consistent meanings? For instance, below is a datapoint taken from the BigBenchHard dataset:
  ```
  Jane quited her job on Mar 20, 2020. 176 days have passed since then. What is the date today in MM/DD/YYYY?
  (A) 09/12/2020
  (B) 11/12/2020
  (C) 12/12/2020
  (D) 09/12/1961
  (E) 09/17/2020
  ```
In this case, computing IICScore per class makes less sense. Could the author provide more insights on this?

Q3. How does $D^2$ Controller work when $k < |c|$?

Q4. $D^2$ Controller measures data similarity in representation space. Did the authors compare different text encoders and see whether / how they affect $D^2$ Controller?

Q5. In my opinion, adding GPT-3 in Section 5.4 could make the analysis stronger.

Q6. The authors report a setting where they combine KATE and $D^2$ Controller, this is interesting. Now, KATE is selecting $k$ different IC examples per test data point, where $k$ is determined by $D^2$ Controller at a dataset level. While I understand the setting, could the authors provide some insights on, is it necessary, or is there a way to dynamically determine the $k$ for every test data point?

Q7. Could the authors provide some insights on why sometimes LLMs fail to benefit from more IC examples? Do stronger LLMs (e.g., gpt-4) suffer less from this?

### Typos and minor stuff
1. There is an extra quotation mark in the 3rd line of Section 5.1, Datasets.
2. DBPedia does not seem to be a good dataset to include in this work, because the longer text, there can be at most one example per class and thus it is not helpful to demonstrate the $D^2$ Controller.
3. In Section 5.4.4, the authors mention that they get better performance with fewer demonstrations. Maybe a more straightforward way to present this is to report (on average) how many tokens their method queries an LLM, and how does that compare to prior work (default $k$).


### Nov 21

I have read the authors response including those answering other reviewers' questions. I appreciate the authors' effort on clarifying things so I'm happy to raise my score a bit. **However, please note, I give 6 -> 8 only because there is no option of 7. I don't think the current version is as mature as 8.** (E.g., the authors have included quite a bit of new experiments during the rebutal period, mainly in the appendices. It may require some efforts to merge some into the main content, with some non-trivial rewriting.)

---

> ### Author Response · Authors · 2023-11-19
> **Response to Reviewer t5tL (part 1)**
>
> Thanks for your appreciation of our useful topic, neat idea, and well-designed method. About the mentioned questions, our answers are as follows:
>
> > **Q1. How does $D^2Controller$ compare to a simple baseline where k is optimized as a hyperparameter using a validation set?**
>
> **A1**: Thanks for your valuable comment. Following your suggestion, we randomly sample some examples (i.e., 100, 200, 300) to construct validation sets as a baseline for choosing k. The results are as follows (note that the results we report are the average performance of ten datasets):
>
> |                     | GPT-2 1.5B | Cerebras-GPT 2.7B | Cerebras-GPT 6.7B | OPT 13B |
> | ------------------- | ---------- | ----------------- | ----------------- | ------- |
> | **Default setting** | 60.0         | 63.0                | 71.7              | 74.5    |
> | **Validation-100**  | 64.9       | 68.3              | 72.6              | 75.8    |
> | **Validation-200**  | 65.4       | 68.5              | 71.8              | 76.1    |
> | **Validation-300**  | 64.9       | 68.3              | 72.6              | 76.4    |
> | $D^2Controller$     | **67.0**         | **69.3**              | **74.0**                | **76.6**    |
>
> Based on these results, we can observe that using these examples does not lead to the optimal choice of k, and almost all of the results are inferior to $D^2Controller$. This further underscores the effectiveness of using IICScore to select a small number of representative examples.
>
> > **Q2: How does $D^2Controller$ work with `classification` tasks where options have no consistent meanings? e.g., BigBenchHard dataset. Could the author provide more insights on this?**
> >
> > ```
> > Jane quited her job on Mar 20, 2020. 176 days have passed since then. What is the date today in MM/DD/YYYY?
> > (A) 09/12/2020
> > (B) 11/12/2020
> > (C) 12/12/2020
> > (D) 09/12/1961
> > (E) 09/17/2020
> > ```
>
> **A2**: Thanks for your inspiring comment.  The BigBenchHard dataset mentioned above is essentially not a classification task but rather a QA (Question-Answering) task. A possible way to work with it is to convert the QA form into classification form. Specifically, we pair the question sentence with each answer and label the pair with correct answer as "yes" and others as "no". Thus, the task is transformed to a 2-class classification task and we can apply our $D^2Controller$ to the dataset.
>
> > **Q3: How does $D^2Controller$ work when k<|c|?**
>
> **A3**: We are sorry that we did not understand exactly the meaning of the above comment. We try answering it:
>
> According to our understanding, |c| denotes the number of examples for a class c. When k < |c|, we can certainly sample k-shot examples from each class. When k > |c|, we can sample all examples from each class. In this paper, all ten of our datasets meet the condition where k < |c|.
>
> > **Q4. $D^2Controller$ measures data similarity in representation space. Did the authors compare different text encoders and see whether / how they affect $D^2Controller$?**
>
> **A4**: Thanks for your valuable feedback. In this paper, we use LLM as a text encoder in $D^2Controller$ to measure data similarity. Following your suggestion, we also tried another two text encoders (i.e., BERT-large and RoBERTa-large). The results are as follows:
>
> |                                    | GPT-2 1.5B | Cerebras-GPT 2.7B | Cerebras-GPT 6.7B | OPT 13B |
> | ---------------------------------- | ---------- | ----------------- | ----------------- | ------- |
> | **$D^2Controller$(BERT-large)**    | 65.8       | 66.5              | 71.8              | 76.6    |
> | **$D^2Controller$(RoBERTa-large)** | 66.0         | 64.6              | 72.8              | **77.4**    |
> | **$D^2Controller$**                | **67.0**         | **69.3**              | **74.0**                | 76.6    |
>
> We can observe that $D^2Controller$(BERT-large) and $D^2Controller$(RoBERTa-large) perform worse than $D^2Controller$ on most of the LLMs (except for OPT 13B), which verifies the superiority of using GPT-architecture LLMs as the text encoder to measure data similarity in representation space.

---

> > ### Author Response · Authors · 2023-11-19
> > **Response to Reviewer t5tL (part 2)**
> >
> > > **Q5. In my opinion, adding GPT-3 in Section 5.4 could make the analysis stronger.**
> >
> > **A5**: Thanks for your precious suggestion. We have reported the results of GPT-3 (175B) in Section 5.4. Based on these results, we can observe that our previous analysis is also applicable to GPT-3. This change has indeed strengthened our analysis, providing further support for our research findings. Thanks for your suggestion again.
> >
> > |                                             | GPT-2 0.3B | GPT-2 0.8B | GPT-2 1.5B | Cerebras-GPT 2.7B | Cerebras-GPT 6.7B | GPT-3 175B |
> > | ------------------------------------------- | :--------- | :--------- | :--------- | :---------------- | :---------------- | :--------- |
> > | **KATE**                                    | 54.1       | 58.7       | 66         | 65.4              | 73.0                | 82.2       |
> > | **KATE + $D^2Controller$**                  | 54.4       | 59.2       | 64.0         | 67.1              | 72.6              | 82.6       |
> > | **Contextual Calibration**                  | 54.9       | 59.3       | 62.2       | 68.3              | 72.4              | 78.9       |
> > | **Contextual Calibration+ $D^2Controller$** | 54.5       | 62.4       | 67.0         | 68.7              | 74.0                | 80.1       |
> >
> > |                         | GPT-2 0.3B | GPT-2 0.8B | GPT-2 1.5B | Cerebras-GPT 2.7B | Cerebras-GPT 6.7B | GPT-3 175B |
> > | ----------------------- | ---------- | ---------- | ---------- | ----------------- | ----------------- | ---------- |
> > | **Random**              | 54.1       | 58.7       | 66.0         | 65.4              | 73                | 81.3       |
> > | **$D^2Controller$-ED**  | 54.4       | 59.2       | 64.0         | 67.1              | 72.6              | 79.1       |
> > | **$D^2Controller$-Cos** | 54.9       | 59.3       | 62.2       | 68.3              | 72.4              | 80.4       |
> > | $D^2Controller$         | 54.5       | 62.4       | 67.0         | 68.7              | 74.0                | 82.0         |
> >
> > |                             | GPT-2 0.3B | GPT-2 0.8B | GPT-2 1.5B | Cerebras-GPT 2.7B | Cerebras-GPT 6.7B | GPT-3 175B |
> > | --------------------------- | ---------- | ---------- | ---------- | ----------------- | ----------------- | ---------- |
> > | **$k_{\max}$-shot setting** | 54.1       | 58.7       | 66.0         | 65.4              | 73.0                | 81.4       |
> > | **$D^2Controller$**         | 54.5       | 62.4       | 67.0         | 68.7              | 74.0                | 82.0         |
> >
> > > **Q6. The authors report a setting where they combine KATE and  $D^2Controller$, this is interesting. Now, KATE is selecting k different IC examples per test data point, where k is determined by $D^2Controller$ at a dataset level. While I understand the setting, could the authors provide some insights on, is it necessary, or is there a way to dynamically determine the k for every test data point?**
> >
> > **A6**: Thanks for your constructive review. It is meaningful but impractical to dynamically determine the k for every test data point. Here we design a possible way to achieve this goal. First, we select m groups of different k-shot samples for each test data point based on similarity, and then use the IICScore method to construct a validation set for the m groups of k-shot samples. Subsequently, we calculate the accuracy of each group of k-shot samples on the validation set, and finally choose the group with the highest accuracy among the m groups to determine the k value.
> >
> > Compared to our method, dynamically selecting k values for each test data would incur a significant computational cost, roughly tens of times more time-consuming. Moreover, as the number of test data points increases, the time cost will also multiply, which is impractical in real-world applications.

---

> > ### Comment · Reviewer_t5tL · 2023-11-20
> > **Clarify**
> >
> > **Q3-followup**:
> >
> > Sorry for the confusion, I should have put $|\mathcal{C}|$ (as defined in Section 2), which to my understanding means the numebr of classes in a specific task, e.g., a 5-way classification task would have $|\mathcal{C}| = 5$.
> >
> > The question was that, for instance, given a 5-way classification task, how does 3-shot IC learning work with the proposed method.
> >
> > Hope this clarifies.

---

> > > ### Author Response · Authors · 2023-11-20
> > > **Thanks for your response**
> > >
> > > We are sorry that there may be a misunderstanding. As we mentioned in Section 2.2, the k indicates the demonstration numbers we sampled for each class, rather than the total number we sampled. For example, in a 5-way classification task using the 3-shot setting, we sample $5\times3=15$ demonstrations in total. Thus, k < $|\mathcal{C}|$  does not affect our method.

---

> > > > ### Comment · Reviewer_t5tL · 2023-11-20
> > > >
> > > > Ah I see, yes you are right. $n$ is the overall number of IC examples. In this work, do you always assume that $n \ge |\mathcal{C}|$? Thanks.

---

> > > > > ### Author Response · Authors · 2023-11-20
> > > > > **Thanks for your response**
> > > > >
> > > > > Yes, you are right. When k = 1, then n = $|\mathcal{C}|$, and when k > 1, then n > $|\mathcal{C}|$.

---

> ### Author Response · Authors · 2023-11-19
> **Response to Reviewer t5tL (part 3)**
>
> > **Q7. Could the authors provide some insights on why sometimes LLMs fail to benefit from more IC examples? Do stronger LLMs (e.g., gpt-4) suffer less from this?**
>
> **A7**: Thanks for your constructive feedback. We speculate that adding an in-context example to a prompt will have two effects: (1) Providing more information to the prompt, resulting in improvement in performance. (2) Causing the distribution of the prompt to become more different from the pre-training corpus of LLMs, leading to difficulty in understanding the prompt and reducing performance. When more IC examples are added, the direction of the change in performance depends on which effect is more influential. For some datasets on some LLMs, the Effect (2) is stronger, leading to performance degradation when more IC examples are included.
>
> Besides, similar to Section 3, we have conducted pilot experiments with the GPT-4 model on five text classification datasets. Due to budgetary constraints, for each dataset, we use five different seeds to test the model's performance in the 1-shot setting, the default setting (4-shot), and $k_{max}$-shot setting. Note that the maximum input length of the GPT-4 model we use is 8192 tokens, so the maximum shot number for SST-5, CR, MPQA, RTE, and CB is 32, 128, 256, 32, and 16. The results are as follows:
>
> | GPT-4                       | SST-5    | CR       | MPQA     | RTE      | CB       |
> | --------------------------- | -------- | -------- | -------- | -------- | -------- |
> | **1-shot setting**          | 45.3±4.4 | 83.7±1.3 | 67.4±1.0 | 82.7±3.0 | 89.3±1.8 |
> | **Default setting(4-shot)** | 45.7±5.0 | 92.2±2.2 | 83.8±0.3 | 89.1±1.4 | 83.9±2.5 |
> | **$k_{\max}$-shot setting** | 43.6±0.8 | 95.9±0.3 | 90.2±1.1 | 88.7±0.6 | 82.7±1.0 |
>
> From the perspective of a general trend, when the input increases from a 1-shot setting to $k_{max}$- shot setting, the accuracy improves on the CR, MPQA, and RTE datasets while declines on the SST-5 and CB datasets. Moreover, in the above table, the RTE dataset achieves the best performance in the default setting, rather than $k_{max}$- shot setting. Thus, increasing the number of demonstrations in stronger LLM like GPT-4 does not necessarily improve performance.
>
> > **Q8: In Section 5.4.4, the authors mention that they get better performance with fewer demonstrations. Maybe a more straightforward way to present this is to report (on average) how many tokens their method queries an LLM, and how does that compare to prior work (default k).**
>
> **A8**: Thanks for your valuable suggestion. We reported the average number of tokens used by three methods (default k, maximum k, and ours) to query LLM:
>
> |                 | GPT-2 1.5B | Cerebras-GPT 2.7B | Cerebras-GPT 6.7B | OPT 13B |
> | --------------- | ---------- | ----------------- | ----------------- | ------- |
> | **Default k**   | 455.49     | 516.87            | 516.87            | 516.87  |
> | **Maximum k**   | 678.29     | 1345.72           | 1345.72           | 1345.72 |
> | $D^2Controller$ | 603.98     | 885.51            | 1187.37           | 725.89  |
>
> Based on these results, we can observe that our method uses fewer tokens to achieve better performance compared to maximum k. Especially on some LLMs, such as Cerebras-GPT 2.7B and OPT-13B, $D^2Controller$ saves almost 30% and 50% tokens.  Meanwhile, although our method uses more tokens compared to the default k, it achieves an average relative improvement of 5.4% on ten datasets.
>
> > **Q9: About typos.**
>
> **A9**: Thanks for your careful review. We have fixed these typos in the revision.

---

> ### Comment · Reviewer_t5tL · 2023-11-21
> **Thank you**
>
> I have read the authors response including those answering other reviewers' questions. I appreciate the authors' effort on clarifying things so I'm happy to raise my score a bit. **However, please note, I give 6 -> 8 only because there is no option of 7. I don't think the current version is as mature as 8.** (E.g., the authors have included quite a bit of new experiments during the rebutal period, mainly in the appendices. It may require some efforts to merge some into the main content, with some non-trivial rewriting.)

---

> ### Author Response · Authors · 2023-11-21
> **Thank you for raising the score**
>
> We are very happy to hear that the reviewer has raised the score! We thank you again for the invaluable feedback and insights. We really appreciate it!

---

### Official Review · Reviewer_mRAG · 2023-11-01

**Soundness:** 2 fair
**Presentation:** 2 fair
**Contribution:** 2 fair
**Rating:** 5
**Confidence:** 4

**Summary:**

The paper proposes an algorithm to select the right number k of examples per class to compose an in-context learning prompt. The proposed Dynamic Demonstration Controller (D2Controller) algorithm chooses k based on a series of experiments with N different in-context learning example selections. In each experiment a validation set is chosen in a special way from the remaining training data that did not make it into the prompt. The k is chosen to maximize the average validation set performance over the N in-context learning support sets.

The paper’s novelty is in the selection of the validation set for each of the N in-context learning support sets. For each of the C classes, the paper chooses an example out of the remaining training data by maximizing a score called IICScore. I did not fully understand the intuition behind the score, but it involves balancing the example’s similarities to the class of interest and to the other classes.

The paper key points are that:
- D2Controller selects k better than the typical settings from the prior work
- D2Controller selects k better than taking as many examples as possible
- D2Controller selects the validation sets better than taking *the same number of validation examples* at random
- D2Controller is also helpful when it is combined with other demonstration selection or ordering methods

**Strengths:**

The paper is most clearly written and methodologically sound. The research question makes sense, the set of baselines is large and appropriate. But there may be one crucial baseline that's missing (see Weaknesses Section).

**Weaknesses:**

This is basically a hyperparameter selection paper, and as such it is missing a key baseline: what if one uses as many examples as possible for selecting k? That would correspond to the classic setting of having your dataset split into training, validation and test sets. While it would be more computationally expensive at the hyperparameter selection time, the key concern in practical applications of LLMs is the inference speed at test time, which would not be affected by using more validation examples to select k.

I imagine that one justification for using fewer validation examples could be that there might be not that many examples available overall. The paper does not discuss this possible constraint though. The set of examples available for selection with IICScore would have to also be restricted.

Some aspects of the paper were difficult to understand, see the next section of the review. I found Figure 5 very dense and difficult to understand. I did not find the motivation for IICScore clearly explained and compelling.

**Questions:**

- “To measure similarities, we transform each sentence x to a vector representation x, which essentially is a language modeling distribution, by querying LLMs with x and obtaining the output” - what does this mean exactly?
- Is your Oracle baseline using the test set examples? If it is, your explanation as to why it is not practical on Page 7 is a bit confusing. Because Oracle would not be a possible practical method, it’s just a hypothetical baseline from above.
- What set of in-context examples is used at the test time? Is it one the N sets you used to select k, or is it another one?

---

> ### Author Response · Authors · 2023-11-19
> **Response to Reviewer mRAG (part 1)**
>
> Thanks for your appreciation of our sound methodology, make-sense questions, and appropriate baselines. We answer your concerns as follows:
>
> > **Q1: What if one uses as many examples as possible for selecting k?**
>
> **A1**: Thanks for your valuable comment. Following your suggestion, we randomly sample more examples as a baseline to select k. Specifically, we construct three different sizes of validation sets (100, 200, and 300) to select k. The results are as follows (note that the results we report are the average performance of ten datasets):
>
> |                     | GPT-2 1.5B | Cerebras-GPT 2.7B | Cerebras-GPT 6.7B | OPT 13B |
> | ------------------- | :--------: | :---------------: | :---------------: | :-----: |
> | **Default setting** |     60.0     |        63.0         |       71.7        |  74.5   |
> | **Validation-100**  |    64.9    |       68.3        |       72.6        |  75.8   |
> | **Validation-200**  |    65.4    |       68.5        |       71.8        |  76.1   |
> | **Validation-300**  |    64.9    |       68.3        |       72.6        |  76.4   |
> | **$D^2Controller$** |     **67.0**     |       **69.3**        |        **74.0**         |  **76.6**   |
>
> Based on these results, we can observe that using more examples does not lead to the optimal choice of k, and almost all of the results are inferior to $D^2Controller$. This further underscores the effectiveness of using IICScore to select a small number of representative examples.
>
> > **Q2: Why use fewer validation examples to select k? Is it because the number of available validation examples is limited?**
>
> **A2**: We are sorry that there may be a misunderstanding, because our method is not proposed for the situation where the number of available examples is limited.
>
> We use fewer validation examples to select k mainly due to the following three reasons:
>
> * **Effective**: As described in A1, using a small number of examples selected by $D^2Controller$ is more effective than using a validation set composed of a larger number of randomly sampled examples.
> * **Efficient**: Our method is much more efficient than using a validation set. In the following Table, we report the running time of selecting k using $D^2Controller$ and using a validation set of 300 examples, respectively. Although the cost of hyperparameter selection would not affect the inference speed at test time, it is still worthwhile accelerating the selection process.
> * **Cheap**: For LLMs that require monetary paid-for API (e.g., GPT-3), it is very expensive to use a large-scale validation set to select hyperparameters, especially for academia. Thus, a method that is effective while requiring less money is necessary.
>
> As for the situation you mentioned in the review, although we do not design our method out of consideration that the available validation examples are limited, we believe our method would be competitive under the situation.
>
> |                       | SST2   | SST5   | MR     | CR     | MPQA   | SubJ   | AGNews | RTE    | CB    |
> | --------------------- | :----- | :----- | :----- | :----- | :----- | :----- | :----- | :----- | :---- |
> | **GPT-2 1.5B**        |        |        |        |        |        |        |        |        |       |
> | $D^2Controller$       | 915s   | 1978s  | 753s   | 654s   | 1112s  | 806s   | 1105s  | 904s   | 1987s |
> | Validation-300    | 5470s  | 4820s  | 4417s  | 3953s  | 6596s  | 4794s  | 3386s  | 5381s  | 2178s |
> | **Cerebras-GPT 2.7B** |        |        |        |        |        |        |        |        |       |
> | $D^2Controller$       | 2360s  | 5386s  | 1946s  | 3654s  | 2778s  | 2096s  | 3242s  | 2419s  | 2694s |
> | Validation-300    | 14700s | 13229s | 11822s | 22346s | 17099s | 12483s | 9771s  | 14572s | 3184s |
> | **Cerebras-GPT 6.7B** |        |        |        |        |        |        |        |        |       |
> | $D^2Controller$       | 4832s  | 10725s | 3942s  | 7076s  | 5558s  | 4223s  | 6432s  | 4773s  | 5376s |
> | Validation-300   | 27653s | 24727s | 23188s | 40582s | 32828s | 25163s | 19100s | 28074s | 6089s |
>
> > **Q3: "*To measure similarities, we transform each sentence x to a vector representation x, which essentially is a language modeling distribution, by querying LLMs with x and obtaining the output*" - what does this mean exactly?**
>
> **A3**: Thanks for your valuable feedback. The intended meaning of this statement is that we input each sentence x into LLMs, thereby obtaining sentence vector representations. We have revised it to make it clear.

---

> > ### Author Response · Authors · 2023-11-19
> > **Response to Reviewer mRAG (part 2)**
> >
> > > **Q4: Is your Oracle baseline using the test set examples? If it is, your explanation as to why it is not practical on Page 7 is a bit confusing. Because Oracle would not be a possible practical method, it’s just a hypothetical baseline from above.**
> >
> > **A4**: Yes, our Oracle baseline uses test set examples. In this paper, the Oracle baseline is actually an approximation to the real upper bound, obtained by iterating through all k-shot settings on the test set. As we mentioned in Section 5.1, to control the GPT-3 inference costs, we randomly sample 256 examples from each dataset to evaluate accuracy. Thus, the iteration is possible on such small-scale of test examples. However, in real-world scenarios, the number of test examples is much larger than 256. Consequently, it becomes impractical to iterate through all k-shot settings to obtain the real Oracle, due to the substantial resource and time requirements.
> >
> > > **Q5: What set of in-context examples is used at the test time? Is it one the N sets you used to select k, or is it another one?**
> >
> > **A5**:  The set of in-context examples at test time is one of the N sets we used to select k.
> >
> > > **Q6: I found Figure 5 very dense and difficult to understand.**
> >
> > **A6**: The leftmost part of Figure 5 shows the iteration process over different k-shot settings (From 1-shot setting to 16-shot setting). The rest of the figure shows an example of how to evaluate the 4-shot setting, which is divided into 3 steps:
> >
> > 1. Sampling $N_s$ groups of 4-shot in-context examples from the training dataset.
> >
> > 2. Sampling evaluation examples from the rest of the training dataset according to IICScore.
> >
> > 3. Computing average prediction accuracy of the selected groups of in-context examples on the evaluation examples.
> >
> > > **Q7: I did not find the motivation for IICScore clearly explained and compelling.**
> >
> > **A7**: Thanks for your valuable feedback on the motivation of IICScore. To save costs, we hope to evaluate the performance of each k-shot setting with a small number of evaluation examples. However, when the number of evaluation examples is small, the evaluation results can be biased. To alleviate the problem, we hope to select a set of evaluation examples that are comprehensive to reflect the performance. Specifically, we co-opt the idea of the fit ability and the generalization ability in ML to achieve comprehensiveness:
> >
> > * The fit ability reflects how well a model can capture seen patterns, and here we adapt the idea by evaluating the performance of a group of in-context examples using similar examples.
> >
> > * The generalization ability is the ability to deal with unseen data, and we analogously use dissimilar examples to evaluate.
> >
> > To measure similarities, considering the type of task is text classification, we propose IICScore which takes both intra and inter-class distances into account to measure similarities.
> >
> > In sum, the motivation behind evaluation examples selection is to construct a set that is small in scale but comprehensive, and IICScore is the metric to help us find these examples.

---

> > > ### Author Response · Authors · 2023-11-22
> > > **Seeking Confirmation on Resolved Concerns and Final Rating with Gratitude**
> > >
> > > Dear Reviewer mRAG,
> > >
> > > We sincerely thank you for the valuable time and effort you have invested in reviewing our paper. As the discussion period draws to a close, with less than 24 hours remaining, we are keen to understand if the concerns you raised in the initial reviews have been effectively addressed. We are open to and appreciate any further feedback, and remain committed to making necessary improvements. We would be grateful if you find that these concerns have been resolved and could reflect this in your final rating of our paper.
> > >
> > > Best, The authors

---

### Author Response · Authors · 2023-11-19
**General Response to Paper Revision**

We thank all reviewers for their insightful and constructive comments. In the following, we will respond to specific concerns raised by different reviewers respectively. Meanwhile, we have updated our submission (with **blue text**) to clarify our approach. For your convenience, we have also listed the changes here:

- In Section 5.4, we have reported the results of GPT-3 (175B) (**Table 2, 3, 4**). This change has indeed strengthened our analysis, providing further support for our research findings.
- In Appendix C, we added the conjectures about what led to the observations in the pilot experiment.
- In Appendix F, we conducted pilot experiments with the GPT-4 model on five text classification datasets and found that increasing the number of demonstrations in stronger LLM GPT-4 does not necessarily improve performance (**Table 8**).
- In Appendix G, we randomly sampled more examples (i.e., 100, 200, 300) as a baseline to select k (**Table 9**).
- In Appendix H, we tried another two text encoders (i.e., BERT-large and RoBERTa-large) to measure data similarity in representation space (**Table 10**).
- In Appendix I, we have reported the average number of tokens used by three methods (default k, maximum k, and **$D^2Controller$** ) to query LLM **(Table 11)**.
- In Appendix J, we provided running times for three different sizes of LLMs during the Evaluation Examples Selection and accuracy-based Evaluation stages, respectively (**Table 12**).
- In Appendix K, we have provided an explanation of the cases in which our method may fail.

Most of the changes are right now in the Appendix for pointing them out more clearly to you. We hope that with these changes we addressed all major concerns. Feel free to leave a comment if you have questions for further discussion, we will try our best to provide satisfactory answers.

---

### Author Response · Authors · 2023-11-21
**Looking forward to further feedbacks**

Dear Reviewers,

Thank you again for your valuable comments and suggestions, which are really helpful for us. We have uploaded new revisions and posted responses to the proposed concerns and questions.

We totally understand that this is a quite busy period of time, since the reviewers may be preparing the rebuttal for their own submissions or rushing for the deadline of the recent conferences.

So we deeply appreciate it if the reviewers can take some time to return further feedbacks on whether our responses and extra experiment results solve your concerns. If there is any other question, please feel free to let us know, we will try our best to provide satisfactory answers.

If your concerns have been addressed, we respectfully ask the reviewer to consider raising the score.

Best,
The authors

---

### Meta-Review · Area_Chair_Ftsf · 2023-12-05

**Metareview:**

The authors present a way to optimize which in context examples optimize performance.
The approach is simple and easy to understand, shows a modest improvement across different LLMs and is more resource friendly than other approaches to optimizing ICL performance. I also appreciate the authors' engagement during discussion. However the overall novelty and performance boost over baselines is modest, and doesn't quite pass the bar for publication. Framing as "best performance with limited resource bandwidth", and why / how the approach suggested achieves this (along with analyses) that would improve the paper.

**Justification For Why Not Higher Score:**

However the overall novelty and performance boost over baselines is modest, and doesn't quite pass the bar for publication.

**Justification For Why Not Lower Score:**

NA

---

### Decision · Program_Chairs · 2024-01-16

Reject